# 1 The impact of tropical cyclones on regional ozone

# 2 pollution and its future trend in the Yangtze River Delta

# 3 of China

- Mengzhu Xi<sup>1</sup>, Min Xie<sup>1</sup>, Da Gao<sup>2, #</sup>, Danyang Ma<sup>1</sup>, Yi Luo<sup>2, 3</sup>, Lingyun Feng<sup>1</sup>,
- Shitong Chen<sup>1</sup>, Shuxian Zhang<sup>1</sup>
- School of Environment, Nanjing Normal University, Nanjing 210023, China
- <sup>2</sup> School of Atmospheric Science, Nanjing University, Nanjing 210023, China
- Ningbo Ecological Environment Monitoring Center of Zhejiang Province, Ningbo 315048, China
- \*\* Now at Center for Aerosol Science and Engineering, Department of Energy, Environmental and
- Chemical Engineering, Washington University in St. Louis, St. Louis, MO, USA
- Correspondence to: Min Xie (minxie@njnu.edu.cn) and Danyang Ma (dyma@njnu.edu.cn)
- **Abstract:** Tropical cyclones (TCs) have a significant impact on ozone (O<sub>3</sub>) in coastal regions by affecting atmospheric circulation and meteorological conditions. This paper investigates the impact and its future 13 changing trends in the Yangtze River Delta (YRD) region. It was found that regional O<sub>3</sub> pollution usually 14 15 occurred before TCs made landfall and after they dissipated in 2018-2022. We classified the weather patterns (SWPs) from June to September during 2018–2022 into four main categories. As pollution levels 16 17 increase within the TCs weather pattern, regional temperatures rise, relative humidity decreases, and 18 wind speeds weaken, creating a favorable environment for O<sub>3</sub> formation and accumulation. The annual 19 O<sub>3</sub> concentration series is reconstructed based on changes in SWP frequency and intensity, quantifying 20 the impact of various SWPs on future O<sub>3</sub> variations. The analysis focuses on the number of days with the TCs weather pattern and their contribution to O<sub>3</sub> variation. Under the SSP2-4.5 and SSP5-8.5 scenarios, 21 22 future YRD O3 concentrations from June to September will increase to varying degrees relative to 23 historical average O<sub>3</sub> concentrations, with average increases of approximately 1.88 μg/m<sup>3</sup> and 6.86 μg/m<sup>3</sup>, 24 respectively. Under all future scenarios, the number of days with TC weather pattern increases to varying
- degrees, and the frequency of TCs increases significantly. The contribution of TCs weather pattern to  $O_3$
- changes is increasing compared to the historical period. This shows that the intensification of climate
- change will intensify the impact of TCs on O<sub>3</sub> in the YRD, and monitoring and early warning need to be
- strengthened.

3132

3637

#### 1 Introduction

In recent years, the concentration of ground-level O<sub>3</sub> in many cities in China has significantly exceeded standards and has intensified. O<sub>3</sub> has replaced Particulate Matter (PM) as the primary pollutant in many regions (Wang et al., 2023b; Yang et al., 2025; Zhao et al., 2020). Ground-level O<sub>3</sub> is mainly generated by photochemical reactions of volatile organic compounds (VOCs) and nitrogen oxides (NO<sub>x</sub>) under sunlight (Lu et al., 2019; Zhang et al., 2024). High concentrations of O<sub>3</sub> can damage the human respiratory system, destroy the human immune system, and even increase the risk of death (Gong et al., 2024; Wang et al., 2025). It can also damage the ecosystem, causing plant leaf necrosis and crop yield reduction (Li et al., 2024). In recent years in the YRD, high emissions of motor vehicle and industrial waste gases, coupled with complex terrain and meteorological conditions due to urbanization, have led to complex air pollution characterized by high concentrations of O<sub>3</sub> and PM (Zhan et al., 2024). According to the 2020 China Ecological Environment Bulletin, in the YRD, the number of days with air

quality exceeding the standard with  $O_3$  as the primary pollutant accounted for over 50% of the total number of days exceeding the standard.

O<sub>3</sub> concentration is regulated by precursor emissions and meteorological conditions during its generation process (Gong et al., 2022; Xie et al., 2016; Xu et al., 2023b). Meteorological factors such as temperature, humidity, precipitation, atmospheric stability, and mixing layer height play important roles in the emission, transportation, and dispersion, chemical reaction, and dry and wet deposition of air pollutants (Chen et al., 2020; Li et al., 2020). Regional O<sub>3</sub> pollution events are often triggered by meteorological conditions such as strong radiation, high temperature, low relative humidity, and low wind speed (Wang et al., 2024b; Zhan and Xie, 2022). Changes in meteorological conditions are affected by weather systems, and the role of weather systems in O<sub>3</sub> concentration changes has also received widespread attention.

Tropical cyclone (TC) activities have a profound impact on the ecological environment of China's coastal areas. In the summer and autumn seasons when O<sub>3</sub> pollution is frequent, TCs are one of the key weather systems that induce O<sub>3</sub> pollution in the YRD (Qi et al., 2024; Shu et al., 2016; Zhan and Xie, 2022). Although the strong winds and precipitation brought by landfalling TCs have a strong scavenging effect on pollutants, the peripheral circulation of TCs away from lands will significantly change the temperature field, wind field, and boundary layer structure, thereby affecting the chemical and physical processes related to O<sub>3</sub> generation, transport, dispersion and deposition, which may in turn aggravate O<sub>3</sub> pollution (Chow et al., 2018; Jiang et al., 2015; Lam et al., 2005; Xu et al., 2023a; Yang et al., 2012). Deng et al. (2019) found that under the influence of the peripheral circulation of TCs, the Pearl River Delta (PRD) is prone to high temperature, low humidity, low wind speed, and strong radiation, which leads to the occurrence of high O<sub>3</sub> and PM concentrations. Hu et al. (2023) focused on analyzing the atmospheric processes that are conducive to the increase in O<sub>3</sub> concentration and the continuous exceedance of O<sub>3</sub> during TC activities, and found that the prevailing downdraft over the PRD brought meteorological conditions of clear sky, low wind speed, high boundary layer, and low relative humidity, which led to the continuous excess of O<sub>3</sub> concentration. Wang et al. (2024a) found that the continuous northward TCs produced and maintained meteorological conditions conducive to the generation of O<sub>3</sub>, promoted the local accumulation and cross-regional transmission of O<sub>3</sub>, and jointly led to a 30 % increase in O<sub>3</sub> concentration in eastern China and a prolonged O<sub>3</sub> pollution period. Xu et al. (2023a) further revealed that when a typhoon landed in the YRD, the contribution of BVOC to O<sub>3</sub> in the Beijing-Tianjin-Hebei region reached 10 ppb; when the TCs moved to Beijing-Tianjin-Hebei region, the cross-transport between the northern China air mass and the YRD contributed half of the O3 related to biological emissions. The peripheral winds and downdrafts of TCs lead to high temperatures and stable weather, which affect O<sub>3</sub> concentrations by affecting regional transport and biological emissions.

The Sixth Assessment Report of the Intergovernmental Panel on Climate Change (IPCC) has established that the rise in the average surface temperature of the earth, caused by human activities, has become a scientific consensus. Compared with the average surface temperature before industrialization (1850–1900), the global average surface temperature increased by about 0.8–1.3 °C due to human activities during 2010–2019 (Adak et al., 2023). In the context of global climate change, with the increases in greenhouse gas emissions, global warming, and interannual climate changes, the state of the atmosphere and ocean has changed, and sea-level rise and extreme climate events have occurred frequently, which have a low-frequency modulating effect on TCs (Chu et al., 2020; Moon et al., 2023; Moon et al., 2025; Wang et al., 2023a). The activity characteristics of TCs have changed significantly, and rare high-intensity TCs have frequently made landfall in China (Wu, 2023). The southeastern coastal

area, as one of the most economically developed and densely populated regions in China, is the region most seriously affected by TCs. Most TCs that affect China originate in the northwest Pacific Ocean, and their movement paths are influenced by the WPSH. Before TCs make landfall from the northwest Pacific Ocean, China's coastal urban agglomerations often experience regional multi-day severe O<sub>3</sub> pollution (Wang et al., 2022b). Although the frequency of TCs in the northwest Pacific Ocean has decreased in recent years, their average intensity has shown an upward trend (Balaguru et al., 2024; Bhatia et al., 2022; Chand et al., 2022; Jung, 2025; Wang et al., 2022a), and their duration has also become longer (Kossin, 2018; Zhang et al., 2020). Yamaguchi and Maeda (2020) showed that although climate warming has accelerated the overall movement speed of TCs, their speed will slow down when they move toward the temperate zone during the poleward migration process, and the time they stay in a specific area will be extended. Affected by the surrounding atmospheric circulation, the slower the TCs move, the longer their impact duration and the greater their effects, which influence O<sub>3</sub> transport, extend the duration of O<sub>3</sub> pollution, exacerbate O<sub>3</sub> concentration, and expand the spatial extent of pollution. Global climate models (GCMs) are able to directly simulate surface O<sub>3</sub> concentrations under both historical and future scenarios (Turnock et al., 2020). These simulations provide an important reference for understanding the long-term evolution of surface O<sub>3</sub> driven by changes in emissions and climate. In the context of global warming in the future, the increase in unfavorable meteorological conditions will make the O<sub>3</sub> pollution problem more serious (Fu and Tai, 2015; Keeble et al., 2017; Arnold et al., 2018; Akritidis et al., 2019; Saunier et al., 2020). Therefore, studying the trend of O<sub>3</sub> changes under future climate change scenarios is particularly important for formulating countermeasures against O<sub>3</sub> pollution.

The YRD is located in a key position in the eastern coastal economic zone of China. It is also a typical area frequently affected by TCs in the warm seasons (Wu et al., 2025; Qi et al., 2024). In addition, it has a high degree of urban agglomeration, dense population, and high energy consumption, making it a key area for air pollution prevention and control (Zhan et al., 2024; Bao et al., 2025). This study focuses on the regulatory mechanism of TCs on regional O<sub>3</sub> pollution and its impact on future pollution trends. First, based on the weather classification over the years, the characteristics of atmospheric circulation changes that cause O<sub>3</sub> changes are revealed, and the mechanism of the TC weather pattern in the YRD affecting O<sub>3</sub> pollution is clarified. Secondly, based on the data reconstruction method, the O<sub>3</sub> annual variation series is reconstructed according to the frequency and intensity of SWPs. Finally, based on future scenario data, the evolution characteristics of atmospheric circulation and the trend in O<sub>3</sub> concentrations under different future scenarios are estimated, and the number of days when TC weather pattern appears and their contribution to O<sub>3</sub> variation are quantified. The research results help to fully and systematically understand the influence mechanism of weather conditions on O<sub>3</sub> pollution, provide a reference for the YRD to carry out targeted O<sub>3</sub> pollution control strategies, and have dual significance in improving regional air quality and advancing low-carbon development.

## 2 Materials and methods

#### 2.1 O<sub>3</sub> observation data

The hourly pollutant monitoring data of 26 cities in the YRD used are derived from the National Environmental Monitoring Center of China. The platform provides pollutant concentration data updated every hour. To better describe the level of O<sub>3</sub> pollution at the urban scale, the arithmetic mean of the pollutant concentrations at each monitoring station was used as the pollutant concentration of the city. O<sub>3</sub>-8h represents the daily maximum 8-hour average O<sub>3</sub> concentration, which can more accurately

characterize the long-term exposure to regional O<sub>3</sub> pollution. The daily maximum 8-hour average O<sub>3</sub> concentration is calculated as the highest average O<sub>3</sub> mixing ratio over any consecutive 8-hour period within a calendar day (00:00–23:59 local time). Specifically, 8-hour moving averages are computed for all possible consecutive 8-hour windows (e.g., 00:00–07:59, 01:00–08:59, ..., 16:00–23:59), and the maximum value among them is recorded as the daily O<sub>3</sub> concentration. Therefore, the O<sub>3</sub> data used in the analysis of this article are all based on the O<sub>3</sub>-8h value, and the daily maximum 8-hour average concentration (unit: μg/m<sup>3</sup>) is taken as the daily O<sub>3</sub> concentration.

## 2.2 Classification of synoptic weather patterns

Common objective weather classification methods mainly include principal component analysis, clustering algorithms, variance method based on correlation coefficients and neural network algorithms. We use orthogonal rotation principal component analysis in T mode (PTT) to classify SWPs in the YRD from 2018 to 2022. The PTT classification method has been integrated into the "cost733class" software. This method is an objective circulation classification method based on principal component analysis. By rotating the loadings of the T-mode principal components, its classification results are more physically interpretable (Philipp et al., 2016). It can accurately reflect the characteristics of the initial circulation field, does not change significantly across different classification objects, and the obtained circulation spatio-temporal field is more stable (Huth et al., 2016). It has been widely used in research fields such as atmospheric circulation and air pollution (Hou et al., 2019; Gao et al., 2021).

The meteorological data are derived from the reanalysis data provided by the National Center for Environmental Prediction (NCEP). The dataset has a horizontal resolution of  $2.5^{\circ} \times 2.5^{\circ}$ , with  $144 \times 73$  grid points in the latitude-longitude domain and 17 vertical layers ranging from 1000 hPa to 10 hPa. Meteorological variables considered in this study include the 500 hPa geopotential height field, sea level pressure, 500 hPa wind field, 850 hPa wind field, 1000 hPa wind field, and vertical wind velocity. Considering that the geopotential height field at 850 hPa effectively minimizes the influence of surface conditions on atmospheric motion while capturing the variations of shallow meteorological systems (Shu et al., 2017). In this study, we used the geopotential height field at 850hPa from June to September during 2018–2022 to classify SWPs and analyze the three-dimensional structure of circulation fields associated with different SWPs.

#### 2.3 Reconstruction of annual variation series of O<sub>3</sub> concentration

The change in SWPs includes the changes in SWP frequency and intensity. The change in SWP frequency refers to the number of occurrences of a certain type of SWP in different years, while the change in SWP intensity refers to the change in the average intensity of the weather system associated with a certain type of SWP across different years. To quantify the contribution of the change in SWP frequency and intensity to the annual change of O<sub>3</sub>, Yarnal (1993) proposed a method to assess the influence of changes in SWP frequency on the annual change of O<sub>3</sub>. The specific equation is as follows:

$$\overline{\overline{O_{3m}}}(fre) = \sum_{k=1}^{6} \overline{O_{3k}} F_{km} \tag{1}$$

where  $\overline{O_{3m}}(fre)$  represents the reconstructed mean  $O_3$  concentration influenced by the frequency variation in SWPs for year m;  $\overline{O_{3k}}$  represents the average  $O_3$  concentration of a certain SWP in all years, and  $F_{km}$  represents the frequency of occurrence of SWP for year m.

Later, Hegarty et al. (2007) proposed that the impact of SWP changes on the annual variation of O<sub>3</sub> should take into account changes in both frequency and intensity. Therefore, Eq. (1) was modified as follows:

$$\overline{\overline{O_{3m}}}(fre+int) = \sum_{k=1}^{6} (\overline{O_{3k}} + \Delta O_{3km}) F_{km}$$
(2)

where  $\overline{O_{3m}}(fre+int)$  represents the reconstructed mean  $O_3$  concentration influenced by the frequency and intensity changes of SWPs for year m;  $\Delta O_{3km}$  represents the modified anomaly concentration value obtained by fitting the SWP k intensity factor in the year m with the  $O_3$  concentration anomaly value ( $\Delta O_3$ ) for that year, which represents the oscillation of the  $O_3$  concentration value caused by the change in the intensity of the SWP k in year m. Hegarty et al. (2007) used the average sea level pressure value in the classification area to represent the SWP intensity change factor. Liu et al. (2019) used the same method to construct the annual  $O_3$  variation series in North China, but used the lowest pressure value in the classification area as the SWP intensity factor.

Since the characteristic changes of each SWP that lead to the increase in  $O_3$  concentration are different, when defining the SWP intensity factor, it is necessary to consider the unique change characteristics that lead to the increase in  $O_3$  concentration for each SWP. To better describe the changes in SWP intensity, we selected the average, maximum, and minimum geopotential heights of different regions based on meteorological characteristics and the location of weather systems. The region with the largest correlation coefficient with the annual  $O_3$  variation series was defined as the SWP intensity factor under that pattern. This part is explained in detail in Section 3.4.1. We select the SWP intensity factor in SWP1 as the maximum geopotential height in zone 7 ( $110^{\circ}-130^{\circ}E$ ,  $20^{\circ}-35^{\circ}N$ ); SWP2 as the maximum geopotential height in zone 9 ( $110^{\circ}-130^{\circ}E$ ,  $20^{\circ}-40^{\circ}N$ ); and SWP4 as the minimum geopotential height in zone 4 ( $110^{\circ}-130^{\circ}E$ ,  $25^{\circ}-40^{\circ}N$ ).

#### 2.4 CMIP6 future climate scenario data

Based on 18 models of the Sixth International Coupled Model Intercomparison Project (CMIP6) and the ERA5 dataset, a bias-corrected global dataset was constructed (Xu et al., 2021). The dataset covers the historical period from 1979 to 2014 and the future climate scenarios from 2015 to 2100, with a horizontal grid spacing of 1.25°×1.25° and a time interval of 6 hours. Considering that O<sub>3</sub> pollution often occurs under extreme weather conditions, the more extreme SSP5-8.5 scenario was selected for this study. This scenario is a high-forcing scenario, and the radiative forcing stabilizes at 8.5 W/m² by 2100. In addition, as China implements more and more energy-saving and emission reduction measures, especially with the establishment of carbon peak and carbon neutrality goals, future greenhouse gas emissions are expected to be effectively controlled. Therefore, the relatively mild SSP2-4.5 scenario was selected for comparison. This scenario is a moderate forcing scenario, and the radiative forcing stabilizes at 4.5 W/m² by 2100. We use the geopotential height field at 850 hPa from June to September during 2018 to 2022 (historical period), 2030 (carbon peak), 2035 (beautiful China), 2060 (carbon neutrality), and 2100 as the input for the PTT to classify weather patterns.

#### 3 Results and discussions

204205

211212

219220

222223

#### 3.1 Characteristics of O<sub>3</sub> pollution in the YRD

Figure 1 shows the monthly variation trend of ground-level O<sub>3</sub> concentrations and meteorological conditions (solar radiation, temperature, precipitation, and relative humidity) in the YRD from 2018 to 2022. It can be seen that the inter-month variations of O<sub>3</sub> concentrations present an M-shaped pattern. From January to June, as the temperature rises and solar radiation increases, the O<sub>3</sub> concentration rises significantly and reaches the first peak (133.36 μg/m<sup>3</sup>). However, the increased precipitation and higher humidity in July exert a notable wet scavenging effect on O<sub>3</sub> precursors. O<sub>3</sub> concentration increases again and reaches the second peak (126.38 µg/m<sup>3</sup>) in September. From October to December, as the temperature gradually decreases and solar radiation weakens, the O<sub>3</sub> concentration shows a continuous downward trend, and reaches the lowest value of 54.17 μg/m<sup>3</sup> of the year in December. The O<sub>3</sub> concentration shows obvious seasonal changes throughout the year. In summer, higher temperatures, stronger solar radiation, and longer sunshine duration jointly enhance atmospheric photochemical reactions, thereby significantly increasing the O<sub>3</sub> concentration. In winter, lower temperatures, weaker solar radiation, and shorter sunshine duration lead to a significant weakening of photochemical reactions, which is not conducive to the generation of O<sub>3</sub>. From the kernel density curve, it can be seen that the distribution in summer months is wider, indicating that the O<sub>3</sub> concentration fluctuates greatly, while the distribution in winter months is narrower, indicating that the O<sub>3</sub> concentration changes are relatively stable in winter. This periodic variation reflects the high sensitivity of O<sub>3</sub> concentration in the YRD to meteorological conditions (temperature, solar radiation, humidity, wind speed, etc.), which affect the rate of photochemical reactions and thus affect the generation of O<sub>3</sub>.

**Figure 1.** Inter-monthly variations in mean O<sub>3</sub> concentrations and associated meteorological conditions in the YRD from 2018 to 2022. (a) The monthly variation of O<sub>3</sub> concentrations is shown, with the red trend line representing the monthly mean values and the individual monthly concentrations labeled. (b-c) The variation trends of solar radiation, temperature, precipitation and relative humidity during the same period. (The shaded area in the violin plot represents the kernel density curve, with the three black lines indicating the 25th percentile, the mean, and the 75th percentile, respectively.)

O<sub>3</sub> pollution in the YRD is concentrated in warm months of the year. We analyzed the temporal distribution of O<sub>3</sub> concentrations in representative cities in the YRD (Shanghai, Nanjing, Hangzhou, and

248249

Hefei) from April to September during 2018-2022 (Fig. 2). Since TCs mainly occur after June, O<sub>3</sub> pollution is often associated with TC activity. During the same TC period, the O<sub>3</sub> concentrations in Shanghai, Nanjing, Hangzhou, and Hefei were different, but the temporal trends were similar. TCs have a certain impact on the changes in O<sub>3</sub> concentrations in the YRD. According to the evolution and trajectory of TC weather, TCs affecting China are primarily from the northwest Pacific Ocean (Zhan et al., 2020; Wang et al., 2024a). As they develop and move, these TCs will have a significant impact on O<sub>3</sub> concentrations in China (Xi et al., 2025). At this time, the YRD is located on the periphery of the TCs. Under the control of the periphery of the TCs, the strong downward airflow will make the YRD in stagnant weather conditions and inhibit the diffusion of pollutants (Shu et al., 2016), accompanied by high temperature, clear and dry weather conditions, which are the main weather conditions causing O<sub>3</sub> pollution. With the evolution of TC weather system, when the TCs center gradually approaches the YRD until it is within a certain range, the YRD is no longer under the influence of the periphery of the TCs and comes under the control of the TC wind and rain belt. Strong winds and precipitation can significantly cleanse air pollutants, and thereby reduce O<sub>3</sub> concentrations. After TCs dissipate, O<sub>3</sub> pollution levels may rise again due to restored meteorological conditions conducive to O<sub>3</sub> formation (Zhan et al., 2020). These results suggest that TCs can significantly affect O<sub>3</sub> pollution in the YRD, highlighting the importance of implementing O<sub>3</sub> control measures prior to TC landfall.

**Figure 2.** Changes in O<sub>3</sub> concentration in Shanghai, Nanjing, Hangzhou, and Hefei from April to September in 2018 to 2022. The gray shadows mark the time of impact of the TCs weather that landed, and the red dots are O<sub>3</sub>-polluted days.

#### 3.2 Main synoptic weather patterns in the YRD

Based on the analysis of O<sub>3</sub> variation characteristics in the YRD, O<sub>3</sub> concentrations were generally high from June to September, coinciding with the peak season of TC activity. We selected June to September during 2018–2022 as the research period and used the PTT weather classification method to classify the weather situation. SWPs in the YRD during this period were primarily divided into four categories. As shown in Table 1, SWP1 is the main SWP, occurring on 383 days and accounting for 62.79 % of the period. SWP2 and SWP4 followed, occurring on 81 and 80 days, respectively, with frequencies of 13.28 % and 13.11 %. SWP3 was less frequent, occurring on 10.82 % of days. Specifically, SWP1 was mainly influenced by the southwesterly flow introduced by the WPSH and the northeast China low, SWP2 by northwesterly flow introduced by a continental high and the northeast China low, SWP3 by southeasterly flow introduced by the WPSH, and SWP4 by northeasterly flow introduced by the WPSH and TCs.

Statistics of average O<sub>3</sub> concentrations and meteorological factors under each SWP indicate that SWP4 had the lowest average O<sub>3</sub> concentration (115.07 μg/m³), whereas SWP2 had the highest value (132.36 μg/m³). O<sub>3</sub> concentrations during SWP1 and SWP3 were similar, at 123.81 μg/m³ and 123.28 μg/m³, respectively. However, significant differences in O<sub>3</sub> concentrations were observed among SWPs during the same period, highlighting the strong influence of circulation patterns on O<sub>3</sub> pollution levels. SWP1 exhibited higher temperatures (27.12 °C), favoring increased O<sub>3</sub> concentrations. SWP2 was influenced by dry northwesterly airflow, exhibiting lower humidity (76.74 %) and slower wind speed (2.03 m/s), which hindered air pollutant dispersion and resulted in higher O<sub>3</sub> concentrations. SWP3 was characterized by lower humidity (78.53 %) and slower wind speed (2.11 m/s), which were key factors contributing to increased O<sub>3</sub> concentration. Under SWP4, though TCs far from the coastline could lead to regional O<sub>3</sub> pollution (in section 3.3.2), the landfalling TCs could bring strong winds and rainstorms, increasing humidity (80.03 %) and wind speed (2.64 m/s), which resulted in lower average O<sub>3</sub> concentrations.

**Table 1.** The occurrence days and frequencies of various SWPs in the YRD from 2018 to 2022, average O<sub>3</sub> concentrations and meteorological factors associated with each SWP.

| Туре | Number of days<br>(frequency) | $O_3$ concentration $(\mu g/m^3)$ | Temperature (°C) | Relative<br>humidity<br>(%) | Wind speed (m/s) |
|------|-------------------------------|-----------------------------------|------------------|-----------------------------|------------------|
| SWP1 | 383(62.79%)                   | 123.81                            | 27.12            | 80.80                       | 2.19             |
| SWP2 | 81(13.28%)                    | 132.36                            | 25.00            | 76.74                       | 2.03             |
| SWP3 | 66(10.82%)                    | 123.28                            | 25.68            | 78.53                       | 2.11             |
| SWP4 | 80(13.11%)                    | 115.07                            | 26.61            | 80.03                       | 2.64             |

Figure 3 illustrates the three-dimensional atmospheric circulation structures of SWP1, SWP2, and SWP3. For SWP1, at 850 hPa, the YRD was located northwest of the WPSH and south of the northeast China low. This influence was controlled by southwesterly flow, which was jointly guided by the WPSH and the northeast China low (Fig. 3a). Warmer southerly winds promoted temperature increases in the YRD, and elevated temperatures enhanced photochemical reactions producing O<sub>3</sub>. At sea level, the

WPSH shifted northeastward, influencing the YRD through southerly flow (Fig. 3d). These southerly winds transported pollutants from the PRD to the YRD. The combined effects of enhanced photochemical reactions and interregional advection contributed to elevated O<sub>3</sub> concentrations in the YRD under SWP1. At 500 hPa, the YRD was dominated by straight westerly flows (Fig. 3g), indicating intensified subsidence.

For SWP2, the YRD was located east of the continental high and southwest of the northeast China low. This influence was driven by northwesterly flow, which was jointly guided by the continental high and the northeast China low (at 850 hPa, Fig. 3b). These dry northwesterly winds significantly reduced relative humidity in the region. At sea level, the YRD was primarily controlled by the continental high (Fig. 3e). At 500 hPa, downward airflow strengthened over the northern YRD behind the trough (Fig. 3h). The significantly lower relative humidity and slower wind speeds favored the formation and accumulation of O<sub>3</sub>, ultimately leading to increased O<sub>3</sub> concentrations under this pattern.

For SWP3, at 850 hPa, the YRD was located southwest of the WPSH and was controlled by southeasterly flow at its southern edge (Fig. 3c). This southeasterly airflow from the ocean increased regional humidity and lowered temperatures. At sea level, the WPSH shifted northward, and the YRD was affected by easterly winds controlled by a tropical depression (Fig. 3f). At 500 hPa, the YRD was located near the edge of the WPSH or close to a low-pressure trough (Fig. 3i). As the WPSH weakened and retreated eastward, its intensity and position suppressed convective activity, leading to a dominant downdraft. This downdraft, combined with lower horizontal wind speeds, hindered the diffusion of pollutants, resulting in increased O<sub>3</sub> concentrations.

**Figure 3.** The average weather conditions in SWP1, SWP2 and SWP3, including an 850 hPa geopotential height field superimposed on wind field (**a-c**), sea level pressure field superimposed on 1000 hPa wind field (**d-f**), a 500

hPa geopotential height field superimposed on wind field (g-i). In (a)-(i), shading represents geopotential height and color vectors represent wind with temperature. The black frame in (a)-(i) includes the YRD.

#### 3.3 The important role of TCs on regional O<sub>3</sub> pollution

#### 3.3.1 The impact of TC weather pattern (SWP4) on O<sub>3</sub> pollution in the YRD

Figure 4 illustrates the three-dimensional atmospheric circulation structure under the TC weather pattern (SWP4). For the TC weather pattern, similar circulation conditions were observed at 850 hPa (Fig. 4a) and at sea level (Fig. 4b). The YRD was located northwest of the TCs and was controlled by northeasterly flow guided by the TCs. The direction and intensity of the northeast wind had a significant impact on meteorological conditions and pollutant transport in the YRD. At 500 hPa, the region was dominated by westerly or northwesterly flow (Fig. 4c). Meanwhile, the peripheral downward airflow associated with lower-level TCs (Fig. 4d) led to a more stagnant atmosphere over the YRD. As the TC approaches the YRD, strong northeasterly flow increased clean sea airflow transportation to the region, lowered temperatures and increased humidity, creating unfavorable meteorological conditions for photochemical reactions. Furthermore, higher wind speeds facilitated air pollutant elimination, leading to a decrease in O<sub>3</sub> concentrations in the region (Table 1).

Figure 4. The average weather conditions in SWP4, including an 850 hPa geopotential height field superimposed on wind field (a), sea level pressure field superimposed on 1000 hPa wind field (b), a 500 hPa geopotential height field superimposed on wind field (c), height-latitude cross-sections of vertical velocity (unit: 10<sup>-2</sup> Pa·s<sup>-1</sup>) between 25°N and 40°N (d). In (a)-(c), shading represents geopotential height and color vectors represent wind with temperature. The black frame in (a)-(c) and the vertical line area in (d) includes the YRD.

# 3.3.2 Changes of atmospheric circulation field with different O<sub>3</sub> pollution levels under TC weather pattern

329

332333

336337

342343

344345

346347

348349

355356

362363

To better explore the relationship between O<sub>3</sub> pollution and the TC weather pattern (SWP4), O<sub>3</sub> pollution was further classified into clean type (C), local pollution type (LP), and regional pollution type (RP) based on pollution characteristics. RP was defined as all days when more than 20 % (> 5) of the 26 cities in the YRD recorded a maximum daily O<sub>3</sub> concentration exceeding 160 µg/m<sup>3</sup>, LP was defined as all days when fewer than 20 % (< 5) of the 26 cities in the YRD recorded a maximum daily O<sub>3</sub> concentration exceeding 160 µg/m<sup>3</sup>, and C included the remaining days not classified as LP or RP. Table 2 presents the number of days and frequencies of C, LP, and RP under SWP4. The frequencies of SWP4 C, SWP4 LP, and SWP4 RP were 56.25 %, 18.75 %, and 25.00 %, respectively. The average O<sub>3</sub> concentrations under these three types were 93.46 µg/m³, 126.97 µg/m³, and 161.19 µg/m³. The clean type occurred more than half the time, contributing to the lower average O<sub>3</sub> concentration (115.07μg/m<sup>3</sup>) under SWP4 (Table 1). Under the TC weather pattern, meteorological factors corresponding to different O<sub>3</sub> pollution characteristics varied significantly (Table 2). As pollution levels increased, regional air temperatures gradually rose, relative humidity decreased, and surface wind speeds decreased significantly. These high temperatures, low humidity, and low wind speeds created favorable conditions for O<sub>3</sub> formation and accumulation. Higher temperatures accelerated photochemical reactions, thereby promoting O<sub>3</sub> production. Low humidity reduced the inhibitory effect of water vapor on photochemical reaction chains, favoring further O<sub>3</sub> accumulation. Reduced wind speeds led to poor atmospheric diffusion, hindering the dilution and transport of precursors and O<sub>3</sub>. Overall, as pollution levels increased, meteorological conditions under the TC weather pattern gradually became more favorable for O<sub>3</sub> formation and accumulation.

**Table 2.** The number of days and frequency of occurrence of clean type (C), local pollution type (LP), and regional pollution type (RP) under each SWP, average O<sub>3</sub> concentrations and meteorological factors associated with each type.

| Туре |         | Number of days<br>(frequency) | $O_3$ concentration $(\mu g/m^3)$ | Temperature (°C) | Relative<br>humidity<br>(%) | Wind speed (m/s) |
|------|---------|-------------------------------|-----------------------------------|------------------|-----------------------------|------------------|
|      | SWP4_C  | 45(56.25%)                    | 93.46                             | 25.57            | 82.06                       | 2.98             |
| SWP4 | SWP4_LP | 15(18.75%)                    | 126.97                            | 27.14            | 79.28                       | 2.65             |
|      | SWP4_RP | 20(25.00%)                    | 161.19                            | 28.54            | 76.03                       | 1.87             |

Figure 5 illustrates the structural characteristics of the three-dimensional atmospheric circulation fields corresponding to different O<sub>3</sub> pollution levels (SWP4\_C, SWP4\_LP, and SWP4\_RP) under SWP4. For SWP4\_C, at 850 hPa, the YRD was controlled by the TCs (Fig. 5a). The TCs brought strong winds and rainstorms to the YRD, which were not conducive to the accumulation of air pollutants or to O<sub>3</sub> formation via photochemical reactions. Simultaneously, high wind speeds and rainfall facilitated pollutant removal. When the TCs made landfall in the YRD (Fig. 5d), significant updrafts occurred in the lower atmosphere, favoring pollutant dispersion. Consequently, clean days were observed in the YRD under the influence of TCs. For the local and regional pollution patterns (SWP4\_LP and SWP4\_RP), TCs were generally 500–1000 km from the YRD coastline (Fig. 5e and f). Previous studies have shown that downdrafts caused by the outer airflow of TCs before landfall led to more stable atmospheric conditions (Zhan et al., 2020; Zhan and Xie, 2022). Under SWP4\_LP and SWP4\_RP, the YRD experienced higher temperatures

and lower humidity (Table 2). These meteorological conditions favored  $O_3$  formation, leading to the occurrence of  $O_3$  pollution events.

For SWP4\_LP, the warm high pressure over North China weakened, and the TC shifted westward, closer to the coastline (Fig. 5e). At 850 hPa, pollutant air masses from the PRD were transported to the YRD, but were diluted by clean air from the ocean (Fig. 5b). Compared to SWP4\_RP, SWP4\_LP had stronger winds and higher relative humidity (Table 2). Consequently, O<sub>3</sub> pollution in the YRD was slightly lower under SWP4\_LP than under SWP4\_RP. For SWP4\_RP, the warm high pressure over North China weakened further, and the TC shifted eastward compared to SWP4\_LP (Fig. 5c). The YRD was dominated by strong downdrafts induced by the peripheral flows of the TC (Fig. 5f). This downdraft reinforced stagnant weather, suppressed convective activity, and hindered pollutant dispersion (Shu et al., 2016; Xi et al., 2025). Combined with meteorological analysis in Table 2, the temperature in the YRD under SWP4\_RP was higher (28.54 °C), the relative humidity was lower (76.03 %), and the wind speed was slower (1.87 m/s), all of which favored photochemical reactions and limited pollutant dispersion. Furthermore, downdrafts induced by the peripheral flows of the TC promoted pollutant accumulation, significantly increasing the frequency of O<sub>3</sub> pollution. In summary, when the TC was located at 130°–135°E and 20°–30°N, the YRD was influenced by its sinking airflow, facilitating the formation of O<sub>3</sub> pollution.

**Figure 5.** The average weather conditions in SWP4\_C (left), SWP4\_LP (middle), and SWP4\_RP (right), including an 850 hPa geopotential height field superimposed on wind field (a-c), sea level pressure field superimposed on 1000 hPa wind field (d-f), a 500 hPa geopotential height field superimposed on wind field (g-i). The shading represents geopotential height, black vectors represent wind and the black frame includes the YRD, and the red asterisk in (d-f) indicates the location where the TCs made landfall.

#### 3.4 To what extent TCs impact regional O<sub>3</sub> pollution in the YRD

# 3.4.1 The role of changes in the intensity and frequency of SWPs in the reconstruction of the annual variation series of O<sub>3</sub>

Different dominant SWPs produced varying near-surface meteorological conditions, which in turn affected atmospheric processes such as O<sub>3</sub> photochemical production, transport, diffusion, and wet and dry deposition. Changes in the frequency and intensity of SWPs were two key factors influencing O<sub>3</sub> concentration variations. We reconstructed the annual time series of O<sub>3</sub> concentrations from June to September between 2018 and 2022 by considering only changes in SWP frequency (fre) and changes in both SWP intensity and frequency (fre+int). In our study, we first removed the influence of emission sources on O<sub>3</sub> concentrations based on the results of Yan et al. (2024). Subsequent analyses were conducted using O<sub>3</sub> concentration series that were free from emission source influences. Using the O<sub>3</sub> trend reconstruction method, we quantified the contribution of SWPs to annual variations in O<sub>3</sub> concentrations from June to September.

Figure 6 illustrates the annual variations of reconstruction  $O_3$  concentrations from June to September during 2018 to 2022. When only changes in SWP frequency were considered, the reconstructed time series was relatively flat and did not adequately capture the variation trend of  $O_3$  concentrations. Therefore, changes in SWP frequency had minimal impact on the annual variation of  $O_3$ . When changes in SWP intensity were considered, the reconstructed series more closely resembled the original annual variation series. Therefore, compared to changes in SWP frequency, changes in SWP intensity contributed more to variations in  $O_3$  concentration. To accurately assess the impact of both SWP frequency and intensity on annual  $O_3$  variation, we quantitatively calculated their contributions. The contribution index was defined as the ratio of the interannual variation amplitude of the reconstructed series to that of the original series, i.e.,  $(O_{3max}$  of the reconstructed series –  $O_{3min}$  of the original series –  $O_{3min}$  of the original series). When only changes in SWP frequency were considered, their contribution to the interannual variation was 10.05%. When changes in SWP intensity were additionally included, the contribution increased to 69.66%. This indicates that, compared with changes in SWP frequency, changes in SWP intensity played a more important role in driving interannual variations in  $O_3$  concentrations.

437

**Figure 6.** The trend of the interannual O<sub>3</sub> concentration time series from June to September during 2018 to 2022 in the YRD. The blue line represents the original interannual O<sub>3</sub> time series, whereas the green and red lines represent the trends of reconstructed O<sub>3</sub> concentrations according to the frequency-only and both frequency and intensity of SWP changes, respectively.

In reconstructing the time series of annual O<sub>3</sub> concentrations, we found that the definition of the SWP intensity factor played a crucial role in the reconstruction. Previous studies reconstructed the annual O<sub>3</sub> concentration series by defining the SWP intensity factor as either the regional mean sea level pressure or the regional minimum pressure (Hegarty et al., 2007; Liu et al., 2019). However, this definition showed poor correlation between the SWP intensity factor and the annual O<sub>3</sub> concentration series for certain SWPs. Therefore, we defined the SWP intensity factor for each SWP based on its specific meteorological characteristics, selecting the maximum, minimum, and mean geopotential heights across nine zones, and evaluated its validity by calculating the correlation coefficient with the annual O<sub>3</sub> variation series (Table 3). For SWP1 and SWP2, the maximum geopotential heights in zone 7 and zone 2 were highly correlated with the annual O<sub>3</sub> variation series. For SWP3 and SWP4, the minimum geopotential heights in zone 9 and zone 4 were highly correlated with the annual O<sub>3</sub> variation series. The maximum geopotential height reflects regional wind speeds, which determine the amount of water vapor transported into the region. Compared with SWP1, SWP2 has a larger weather system scale, so the maximum geopotential height in zone 2 shows a stronger correlation with the O<sub>3</sub> series than that in zone 7. For SWP4, the YRD was affected by TCs, and O<sub>3</sub> concentrations were closely related to TC intensity. The minimum geopotential height in zone 4 reflects the TC intensity. When the SWP intensity factor was defined based on the unique meteorological characteristics of each SWP, the reconstructed series more accurately reflected the impact of changes in SWP intensity on O<sub>3</sub> concentrations.

Table 3. Correlation coefficients between the O<sub>3</sub> concentration time series and different SWP intensity factors under each SWP.

| Zone 1 |                        |       |        | Zone 2                 |                        |       | Zone 3                 |                        |       |  |
|--------|------------------------|-------|--------|------------------------|------------------------|-------|------------------------|------------------------|-------|--|
| Type   | (115°~135°E, 20°~40°N) |       | (90°~1 | (90°~140°E, 20°~50°N)  |                        |       | (110°~130°E, 10°~40°N) |                        |       |  |
|        | Mean                   | Max   | Min    | Mean                   | Max                    | Min   | Mean                   | Max                    | Min   |  |
| SWP1   | -0.62                  | -0.72 | -0.41  | -0.69                  | 0.26                   | 0.45  | -0.59                  | -0.76                  | 0.17  |  |
| SWP2   | -0.65                  | -0.87 | -0.45  | -0.47                  | -0.88                  | -0.09 | -0.60                  | -0.84                  | -0.53 |  |
| SWP3   | -0.41                  | -0.67 | 0.72   | 0.10                   | -0.17                  | 0.43  | -0.11                  | -0.60                  | 0.75  |  |
| SWP4   | 0.15                   | -0.32 | 0.36   | 0.26                   | 0.01                   | 0.32  | 0.33                   | -0.38                  | 0.49  |  |
|        | Zone 4                 |       |        |                        | Zone 5                 |       |                        | Zone 6                 |       |  |
| Type   | (110°~130°E, 25°~40°N) |       |        | (100°~                 | (100°~120°E, 15°~35°N) |       |                        | (110°~120°E, 15°~35°N) |       |  |
| ·      | Mean                   | Max   | Min    | Mean                   | Max                    | Min   | Mean                   | Max                    | Min   |  |
| SWP1   | -0.67                  | -0.80 | -0.02  | -0.33                  | 0.37                   | 0.05  | -0.60                  | -0.82                  | 0.10  |  |
| SWP2   | -0.47                  | -0.73 | -0.44  | -0.36                  | -0.59                  | -0.19 | -0.48                  | -0.65                  | -0.29 |  |
| SWP3   | -0.19                  | -0.60 | 0.46   | 0.34                   | 0.31                   | 0.67  | 0.29                   | -0.29                  | 0.85  |  |
| SWP4   | 0.31                   | -0.32 | 0.64   | 0.48                   | -0.02                  | 0.55  | 0.47                   | 0.26                   | 0.55  |  |
|        | Zone 7                 |       |        | Zone 8                 |                        |       | Zone 9                 |                        |       |  |
| Type   | (110°~130°E, 20°~35°N) |       | (115°~ | (115°~135°E, 30°~50°N) |                        |       | (110°~130°E, 20°~40°N) |                        |       |  |
|        | Mean                   | Max   | Min    | Mean                   | Max                    | Min   | Mean                   | Max                    | Min   |  |
| SWP1   | -0.53                  | -0.83 | 0.06   | -0.40                  | -0.73                  | 0.42  | -0.61                  | -0.76                  | 0.09  |  |
| SWP2   | -0.54                  | -0.79 | -0.25  | -0.31                  | -0.72                  | 0.03  | -0.56                  | -0.77                  | -0.50 |  |
| SWP3   | 0.02                   | -0.69 | 0.86   | -0.25                  | -0.64                  | 0.33  | -0.08                  | -0.60                  | 0.87  |  |
| SWP4   | 0.50                   | -0.21 | 0.49   | -0.19                  | -0.03                  | -0.15 | 0.39                   | -0.30                  | 0.48  |  |

## 3.4.2 Changes in TCs and the effects on O<sub>3</sub> concentration over the YRD in the future

After clarifying the relationship between SWPs and O<sub>3</sub> pollution, future trends of O<sub>3</sub> concentrations under different scenarios were estimated based on projected changes in SWPs. Due to its unique geographical location, the YRD was subject to relatively complex weather systems. The PTT method was used to classify the weather conditions in the YRD from June to September under the SSP2-4.5 and SSP5-8.5 scenarios into four main categories. SWP1 was mainly controlled by southwesterly flow associated with the WPSH and the northeast China low, SWP2 by northwesterly flow associated with a continental high and the northeast China low, SWP3 by southeasterly flow associated with the WPSH, and SWP4 by northeasterly flow associated with the WPSH and TCs. Figure 7 illustrates the distribution of days for each SWP from June to September in the YRD during the historical period and under the SSP2-4.5 and SSP5-8.5 future scenarios. According to the average O<sub>3</sub> concentration of each SWP during the historical period (Table 1), SWP1 and SWP2 with higher O<sub>3</sub> concentrations are classified as high-average O<sub>3</sub> patterns. The average annual number of days during the historical period was 92.8. The high-average O<sub>3</sub> pattern occurred most frequently in 2022, with 98 days and higher O<sub>3</sub> concentrations. This confirms that changes in SWP frequency, which characterize specific O<sub>3</sub> concentration patterns, could influence O<sub>3</sub> concentration trends to a certain extent (Hegarty et al., 2007).

Under the SSP2-4.5 scenario, the frequencies of SWP1 to SWP4 were 57.79%, 19.67%, 6.76%, and 15.78%, respectively (Fig. 10). High-average O<sub>3</sub> pattern occurred on 93, 99, 96, and 90 days in 2030, 2035, 2060, and 2100, respectively, showing a general increasing trend, with the peak occurring in 2035. These results suggest that, considering only changes in SWP frequency, O<sub>3</sub> concentrations would reach

a maximum in 2035 under the SSP2-4.5 scenario. Under the SSP5-8.5 scenario, the frequencies of SWP1 to SWP4 were 50.82%, 23.98%, 7.17%, and 18.03%, respectively (Fig. 10). High-average O<sub>3</sub> pattern will occur on 99, 90, 82, and 94 days in 2030, 2035, 2060, and 2100, respectively, under the SSP5-8.5 scenario, with the number of days increasing in 2030 and 2100. This also suggests that, considering only changes in SWP frequency, O<sub>3</sub> concentrations would increase in 2030 and 2100 under the SSP5-8.5 scenario.

**Figure 7.** Distribution of the number of days of occurrence of each SWP in the YRD from June to September under the historical period (2018-2022), SSP2-4.5, and SSP5-8.5 future scenarios.

Due to the frequent occurrence of TCs in the YRD from June to September, they not only brought extreme weather events but also had a significant impact on regional O<sub>3</sub> pollution. We further examined changes in the number of days under the TC weather pattern (SWP4). During the historical period (2018–2022), the annual average number of days with TC weather pattern in the YRD was 16 (Fig. 8). Under the SSP2-4.5 future scenario, this number increased to 19, 17, 21, and 20 days in 2030, 2035, 2060, and 2100, respectively. This indicates that the number of days with TC weather pattern increased to varying degrees, with the lowest number of days in 2035. Under the SSP5-8.5 future scenario, this number increased to 18, 19, 30, and 24 days in 2030, 2035, 2060, and 2100, respectively. Similarly, under the SSP 5-8.5 future scenario, the number of days with the TC weather pattern in the YRD increased to varying degrees. Overall, the frequency of TCs affecting O<sub>3</sub> concentrations in the YRD increased significantly under both future scenarios.

**Figure 8.** Occurrence days of TCs SWP under the historical period, SSP2-4.5 and SSP5-8.5 future scenarios in the YRD from June to September.

#### 3.4.3 Reconstruction of annual O<sub>3</sub> variability in the YRD under future scenarios

Based on the SWP classification results and the reconstructed empirical relationships, statistical projections of the annual O<sub>3</sub> concentration series from June to September were made for the SSP2-4.5 and SSP5-8.5 scenarios. Figure 9 compares the reconstructed annual O<sub>3</sub> concentration series from June to September with the historical period. When considering only changes in SWP frequency, the reconstructed O<sub>3</sub> series under the SSP2-4.5 scenario peaked in 2035. The reconstructed O<sub>3</sub> series under the SSP5-8.5 scenario increased in 2030 and 2100, consistent with the conclusions drawn in Section 3.4.2. However, overall, the reconstructed series from 2018 to 2022 indicated that changes in SWP frequency had little impact on the annual O<sub>3</sub> variation, and the reconstruction could not accurately capture the actual trend. After incorporating changes in the SWP intensity factor, the reconstructed annual O<sub>3</sub> series more closely reflected the actual trend.

The average O<sub>3</sub> concentration in the YRD from June to September during 2018 to 2022 was 123.89 μg/m³. Under the SSP2-4.5 scenario, O<sub>3</sub> concentrations in the YRD are projected to increase relative to the historical period (Fig. 9a), with an average increase of approximately 1.88 μg/m³. Based on the reconstructed contribution of SWP changes, future O<sub>3</sub> concentrations in the YRD are estimated to be 2.70 μg/m³ higher than in the historical period. Under the SSP5-8.5 scenario, O<sub>3</sub> concentrations are projected to increase relative to the historical period (Fig. 9b), reaching 133.80 μg/m³ in 2100, an increase of approximately 6.86 μg/m³. Based on the reconstructed contribution of SWP changes, future O<sub>3</sub> concentrations in the YRD are estimated to be 9.85 μg/m³ higher than in the historical period. In summary, under both the SSP2-4.5 and SSP5-8.5 future climate scenarios, O<sub>3</sub> concentrations in the YRD are projected to increase from June to September, with more severe O<sub>3</sub> pollution under the SSP5-8.5 scenario. Previous studies based on CMIP6 multi-model simulations have shown that surface O<sub>3</sub> concentrations are projected to decrease in response to reductions in anthropogenic emissions, although the magnitude and spatial distribution of changes vary among scenarios (Turnock et al., 2020; Li et al., 2023). The

trends revealed in this study are generally consistent with those of previous studies, lending confidence to the robustness of our findings.

**Figure 9.** The trend of the interannual O<sub>3</sub> concentration time series under SSP2-4.5 (a) and SSP5-8.5 (b) future scenarios in the YRD. The blue line represents the original interannual O<sub>3</sub> time series, whereas the green and red lines represent the trends of reconstructed O<sub>3</sub> concentrations according to the frequency-only and both frequency and intensity of SWP changes, respectively.

Figure 10a shows the contribution of each SWP to the reconstructed series of future O<sub>3</sub> concentrations in the YRD from June to September under the SSP2-4.5 and SSP5-8.5 scenarios. We focus on the contribution of the TC weather pattern (SWP4) to the reconstructed O<sub>3</sub> concentration series under different scenarios (Fig. 10b). During the historical period (2018–2022), the TC weather pattern contributed an average of 13.11% to O<sub>3</sub> concentrations in the YRD. Under the SSP2-4.5 scenario, the contributions of the TC weather pattern in 2030, 2035, 2060, and 2100 were 15.57%, 13.93%, 17.21%, and 16.39%, respectively. The frequency of the TC weather pattern in the YRD in 2035 was lower, resulting in a lower contribution of the TC weather pattern to O<sub>3</sub> concentrations in that year. Under the SSP5-8.5 scenario, the TC weather pattern contributes 14.75%, 15.57%, 24.59%, and 19.67% to O<sub>3</sub> changes in 2030, 2035, 2060, and 2100, respectively. Under both the SSP2-4.5 and SSP5-8.5 scenarios, the contribution of the TC weather pattern to O<sub>3</sub> pollution increases to varying degrees compared with the historical period. This suggests that under future climate change, the impact of TC weather pattern on O<sub>3</sub> pollution in the YRD may intensify. Further research is needed on the relationship between key SWPs, such as TC weather pattern, and O<sub>3</sub> formation mechanisms to more accurately predict and mitigate regional O<sub>3</sub> pollution under future climate conditions.

**Figure 10.** Contribution of SWP to the projected annual variation series of O<sub>3</sub> concentration under the SSP2-4.5 and SSP5-8.5 future scenarios. (a) Contribution of four SWPs, (b) Contribution of TC weather pattern (SWP4).

#### 4 Conclusions

This study investigates the impact of meteorological conditions on O<sub>3</sub> pollution and future trends in the YRD, with a particular focus on the influence and mechanisms of TCs on O<sub>3</sub> pollution. The spatial distribution and temporal variation of O<sub>3</sub> pollution during TC events in the YRD from 2018 to 2022 were analyzed. The PTT objective classification method was applied to identify the main SWPs in the YRD from June to September 2018–2022 and assess their influence on varying levels of O<sub>3</sub> pollution. Finally, using the reconstructed empirical relationship, the annual variation of O<sub>3</sub> concentrations in the YRD from June to September under future scenarios was statistically projected, and the contribution of TCs to future O<sub>3</sub> changes was quantified. The main findings are summarized as follows.

Analysis of O<sub>3</sub> pollution characteristics in the YRD from 2018 to 2022 indicates that inter-month variations of O<sub>3</sub> concentrations exhibit an M-shaped pattern. Peaks generally occur in June and September, with O<sub>3</sub> concentrations showing substantial fluctuations during summer. During TC activity from 2018 to 2022, O<sub>3</sub> pollution in typical YRD cities exhibited similar temporal trends, generally increasing initially and then decreasing, closely associated with the trajectory and center position of the TCs. When the TC center approaches the YRD, regional O<sub>3</sub> concentrations decrease due to strong winds and rainfall, facilitating atmospheric pollutant removal. Conversely, when TCs are in their formation or dissipation stages, or their peripheral airflows influence the YRD, high temperatures, clear skies, and dry conditions favor O<sub>3</sub> formation and accumulation.

Using the PTT objective classification method, four main SWPs were identified in the YRD from June to September 2018–2022. SWP1 is mainly controlled by the southwesterly flow introduced by the WPSH and the northeast China low, SWP2 by northwesterly flow introduced by a continental high and the northeast China low, SWP3 by southeasterly flow introduced by the WPSH, and SWP4 by northeasterly flow introduced by the WPSH and TCs. SWP1 is the dominant SWP, occurring 62.79 % of the time, whereas the TC weather pattern (SWP4) occurs 13.11 % of the time. SWP2 is influenced by dry northwesterly flow, exhibiting lower humidity (76.74 %) and slower wind speeds (2.03 m/s), which hinder pollutant dispersion and lead to higher O<sub>3</sub> concentrations. Under SWP4, TCs bring strong winds and rainstorms, increase humidity (80.03%), and enhance wind speeds (2.64 m/s), resulting in lower average O<sub>3</sub> concentrations.

To investigate the interaction mechanisms between TC weather patterns and O3 pollution, three

#### manuscript submitted to Journal of Atmospheric Chemistry and Physics

 pollution levels, clean type (C), local pollution (LP), and regional pollution (RP), were defined based on  $O_3$  characteristics, and the corresponding atmospheric circulation features were analyzed in detail. For SWP4\_C, TCs make landfall in the YRD, bringing strong winds and rainstorms that inhibit  $O_3$  accumulation and maintain clean air conditions. In contrast, under the SWP4\_LP and SWP4\_RP, downdrafts induced by the peripheral airflows of TCs that have not yet made landfall led to more stagnant atmospheric conditions. Elevated temperatures and weak winds over the YRD create conditions highly favorable for  $O_3$  formation. The mean  $O_3$  concentrations for these three categories were 93.46  $\mu$ g/m³, 126.97  $\mu$ g/m³, and 161.19  $\mu$ g/m³. The clean type occurred more than half the time, contributing to the lower mean  $O_3$  concentration (115.07  $\mu$ g/m³) under the SWP4. When the TC centers are located at 130°–135°E and 20°–30°N, the YRD is influenced by downdrafts from the peripheral flows of the TCs, facilitating the formation of high-concentration  $O_3$  pollution.

The SWP intensity and frequency contribute 10.05% and 69.66% to the annual  $O_3$  variation series, respectively. Compared to frequency, intensity variations have a more significant impact on  $O_3$  variation. Classification of SWPs from June to September under the SSP2-4.5 and SSP5-8.5 future scenarios shows that the number of days with TC weather pattern has increased to varying degrees, and the duration of TCs impacts will increase in the future. SWP intensity factors are defined based on the meteorological characteristics and location of each SWP, and a statistical estimate of the  $O_3$  concentration variation series from June to September in the YRD under future scenarios is made using data reconstruction methods. Under the SSP2-4.5 and SSP5-8.5 future scenarios,  $O_3$  concentrations increase by an average of 1.88  $\mu g/m^3$  and 6.86  $\mu g/m^3$  compared to the historical period. Based on their reduction contributions, the increase is expected to be 2.70  $\mu g/m^3$  and 9.85  $\mu g/m^3$ . Under the two future scenarios, the contribution of TC weather pattern to  $O_3$  pollution has increased to varying degrees compared with the historical period, indicating that the impact of TC weather pattern on  $O_3$  pollution in the YRD may be further intensified in the future.

In summary, O<sub>3</sub> pollution in the YRD is on the rise, and summer O<sub>3</sub> pollution is often related to TC activities, and often occurs before and after TC activities. Through the research of this article, we have deepened our understanding of the mechanism of TCs on regional O<sub>3</sub> pollution. In addition, under the background of global warming, the intensity and duration of TC generation will increase, which will have a serious impact on China's coastal areas. Compared with the direct damage caused by landfalling TCs, the secondary disasters caused by them, such as O<sub>3</sub> pollution, should also be taken seriously. The research results have important scientific significance and practical application value for the in-depth understanding of the formation mechanism of O<sub>3</sub> pollution in the YRD, formulating targeted pollution prevention and control strategies, and improving regional air quality.

## manuscript submitted to Journal of Atmospheric Chemistry and Physics

| 594 | Data availability. Chinese O <sub>3</sub> monitoring data are available at <a href="http://www.cnemc.cn/en/">http://www.cnemc.cn/en/</a> . Meteorological                                        |  |  |  |  |
|-----|--------------------------------------------------------------------------------------------------------------------------------------------------------------------------------------------------|--|--|--|--|
| 595 | data are available at $\underline{\text{http://rda.ucar.edu/datasets/ds083.2/}}. \text{ The TCs trajectory dataset are available at } \underline{\text{http://rda.ucar.edu/datasets/ds083.2/}}.$ |  |  |  |  |
| 596 | https://www.typhoon.org.cn/. CMIP6 future climate scenario data are available at                                                                                                                 |  |  |  |  |
| 597 | https://www.scidb.cn/detail?dataSetId=791587189614968832.                                                                                                                                        |  |  |  |  |
| 598 |                                                                                                                                                                                                  |  |  |  |  |
| 599 | Author contributions. MiX had the original idea for the study, designed the experiments. MeX, DG, and                                                                                            |  |  |  |  |
| 600 | DM collected and processed the data, designed the research and prepared the original draft. MiX discuss                                                                                          |  |  |  |  |
| 601 | the results and revise the original draft and also provided financial support for the project leading to this                                                                                    |  |  |  |  |
| 602 | publication. YL revised the paper and helped to collect the data. LF, SC, and SZ checked the English of                                                                                          |  |  |  |  |
| 603 | the original paper.                                                                                                                                                                              |  |  |  |  |
| 604 |                                                                                                                                                                                                  |  |  |  |  |
| 605 | Competing interests. The contact author has declared that none of the authors has any competing interests.                                                                                       |  |  |  |  |
| 606 |                                                                                                                                                                                                  |  |  |  |  |
| 607 | Financial support. This research has been supported by the National Nature Science Foundation of China                                                                                           |  |  |  |  |
| 608 | (42275102), the Special Science and Technology Innovation Program for Carbon Peak and Carbon                                                                                                     |  |  |  |  |
| 609 | Neutralization of Jiangsu Province (BE2022612), the National Key Basic Research Development                                                                                                      |  |  |  |  |
| 610 | Program of China (2024YFC3711905) and the research start-up fund for the introduction of talents from                                                                                            |  |  |  |  |
| 611 | Nanjing Normal University (184080H201B57).                                                                                                                                                       |  |  |  |  |
| 612 |                                                                                                                                                                                                  |  |  |  |  |

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
