# Peer review of "The impact of tropical cyclones on regional ozone"

_EGUsphere, 2025_

## Author Comment (AC1)

**Response to Reviewers**

| | |
|---|---|
| *Journal:* | *Atmospheric Chemistry and Physics* |
| *Manuscript No.:* | egusphere-2025-2466 |
| *Title of Paper:* | The impact of tropical cyclones on regional ozone pollution and its future trend in the Yangtze River Delta of China |
| *Authors:* | Mengzhu Xi, Min Xie*, Da Gao, Danyang Ma, Yi Luo, Lingyun Feng, Shitong Chen, Shuxian Zhang |

Dear Editor,

We would like to submit our revised manuscript entitled "**The impact of tropical cyclones on regional ozone pollution and its future trend in the Yangtze River Delta of China**" to *Atmospheric Chemistry and Physics*.

On behalf of my co-authors, we thank you for handling the peer review of our manuscript. We appreciate the time and effort put in by you and the referees in reviewing our manuscript. These constructive comments are all valuable for revising and improving our manuscript. We have carefully studied these comments and made correction as requested. In the point-by-point responses, the reviewer's comments are in black, the author's responses are in blue, and the changes made to the text are highlighted in red. The line numbers in the author's responses are obtained from the clean version of revised manuscript, in which all the revisions have been accepted.

Best regards,

Authors

**Anonymous referee #2:**

General comments: The manuscript investigates the impact of tropical cyclones on ozone and its future changing trends in the Yangtze River Delta region. Their results shows that the regional $O_3$ pollution are affected by tropical cyclones, and based on future scenario data, they further discuss the changes in tropical cyclones and their impact on the trend of $O_3$. The results are interesting for understanding the future change of tropical cyclones and $O_3$. Several points of the manuscript still need to be improved before accepted. Specifically, they show that the regional $O_3$ pollution usually occurred before and after tropical cyclones in 2018−2022 as shown in their analysis, why do they use the data reconstruction about the TC weather patterns to evaluate the annual change of $O_3$ concentrations in the future, instead of extreme $O_3$ pollution? I believe that the evaluation of extreme $O_3$ pollution may be reasonable. Therefore, the manuscript needs to make major revisions before their paper is considered acceptable. Please see the following comments.

**Response:** We thank the referee #2 for the constructive comments and suggestions, which are very helpful for improving the clarity and reliability of the manuscript. We have revised some sentences for better readability and enhanced the clarity of some figures. Please see our point-by-point responses to your comments below.

**Main comments:**

1. As show in manuscript, the regional $O_3$ pollution usually occurred before and after tropical cyclones in 2018-2022 as shown in their analysis. They use the data reconstruction about the TC weather patterns to evaluate the annual change of $O_3$ concentrations in the future, instead of extreme $O_3$ pollution? I believe that the evaluation of extreme $O_3$ pollution may be reasonable. Please discuss this point.

**Response:** Thank you for your careful review of our manuscript and your scientific suggestions. Based on your suggestion, we have adjusted the study period. Our previous analysis shows that $O_3$ pollution in the Yangtze River Delta YRD typically occurs during **June–September**. This period not only coincides with the peak tropical cyclone (TC) activity but also corresponds to the highest $O_3$ pollution occurrence, making it the

most relevant timeframe for our study. To better investigate the impact of TCs on $O_3$ pollution, we no longer perform weather pattern classification for the entire year, but focus on June–September each year. After classifying the weather patterns, we found that the frequency and intensity of these patterns are strongly correlated with $O_3$ concentration variations. Following previous studies, we reconstructed the annual warm season $O_3$ variation sequence based on the frequency and intensity of the weather patterns. **In this study, the reconstructed sequence also represents $O_3$ variations during June–September each year when extreme $O_3$ episodes occur.**

We note that tropical cyclone events can lead to extreme $O_3$ episodes. However, as our reconstruction method is based on weather pattern frequency and intensity, it does not allow for analysis of individual extreme events, although it effectively captures the main interannual variability of $O_3$. In this study, we use this method to reconstruct future annual $O_3$ variation sequences in warm season when extreme $O_3$ episodes usually occur, and to explore the changes in the occurrence days of TC-related weather patterns and their contributions to the reconstructed sequences.

In our future work, we will combine numerical simulations with observational data to further investigate the relationship between tropical cyclones and extreme $O_3$ pollution, providing a more comprehensive understanding of the impacts of TCs on $O_3$. Now, the study is going on, and some of the reviewers' suggestions are also adopted. Changes in the revised manuscript (lines 260–263 and 397–406) are shown as follows:

3.2 Main synoptic weather patterns in the YRD

"Based on the analysis of $O_3$ variation characteristics in the YRD, $O_3$ concentrations were generally high from **June to September**, coinciding with the peak season of TC activity. We selected June to September during 2018−2022 as the research period and used the PTT weather classification method to classify the weather situation…"

3.4.1 The role of changes in the intensity and frequency of SWPs in the reconstruction of the annual variation series of $O_3$

"Different dominant SWPs produced varying near-surface meteorological conditions, which in turn affected atmospheric processes such as $O_3$ photochemical

production, transport, diffusion, and wet and dry deposition. Changes in the frequency and intensity of SWPs were two key factors influencing $O_3$ concentration variations. We reconstructed the annual time series of $O_3$ concentrations from **June to September** between 2018 and 2022 by considering only changes in SWP frequency (fre) and changes in both SWP intensity and frequency (fre+int). In our study, we first removed the influence of emission sources on $O_3$ concentrations based on the results of Yan et al. (2024). Subsequent analyses were conducted using $O_3$ concentration series that were free from emission source influences. Using the $O_3$ trend reconstruction method, we quantified the contribution of SWPs to annual variations in $O_3$ concentrations from June to September."

2. The authors utilize the data reconstruction about the TC weather patterns to evaluate the annual change of $O_3$ concentrations in the future. This data reconstruction of $O_3$ concentrations only considers the effects of weather patterns, I want to know the effect from the perturbations in precursor emissions. Maybe the perturbations in precursor emissions can dominate the future change of $O_3$ concentrations. Therefore, please check the relative contributions of climate change (or weather patterns) and perturbations in precursor emissions to future $O_3$ concentrations.

**Response:** Thank you for your helpful suggestion. On the early stage when establishing the reconstruction relationship for weather patterns, we did not remove the influence of emissions. Considering the reviewer's comment, in the revised manuscript, we first applied detrending and **removed the effect of emission sources** from the $O_3$ data before establishing the reconstruction relationships. This allows us to analyze the future impact of weather patterns on $O_3$ more accurately.

We acknowledge that perturbations in precursor emissions may dominate future $O_3$ changes, but evaluating their relative contribution is beyond the scope of the current study. The primary focus of this work is to investigate the role of tropical cyclone-related weather patterns in driving interannual $O_3$ variability. Numerous studies have already analyzed the contribution of emissions to $O_3$, while our study emphasizes the influence of weather patterns on future $O_3$ trends. The reconstruction method used in

this study is also based solely on the frequency and intensity variations of weather patterns and does not include any calculation related to precursor emissions.

In future work, we will combine numerical simulations with emission scenario analyses to further quantify the relative contributions of climate/weather pattern changes and perturbations in precursor emissions to future $O_3$ variations.

Changes in the revised manuscript (lines 397–406) are shown as follows:

3.4.1 The role of changes in the intensity and frequency of SWPs in the reconstruction of the annual variation series of $O_3$

"Different dominant SWPs produced varying near-surface meteorological conditions, which in turn affected atmospheric processes such as $O_3$ photochemical production, transport, diffusion, and wet and dry deposition. Changes in the frequency and intensity of SWPs were two key factors influencing $O_3$ concentration variations. We reconstructed the annual time series of $O_3$ concentrations from June to September between 2018 and 2022 by considering only changes in SWP frequency (fre) and changes in both SWP intensity and frequency (fre+int). In our study, we first removed the influence of emission sources on $O_3$ concentrations based on the results of Yan et al. (2024). Subsequent analyses were conducted using $O_3$ concentration series that were free from emission source influences. Using the $O_3$ trend reconstruction method, we quantified the contribution of SWPs to annual variations in $O_3$ concentrations from June to September."

3. In the global climate models, they can directly simulate the surface $O_3$ concentrations in historical and future scenarios (e.g., Turnock et al., 2020). I suggest that the authors added the results from the Coupled Model Intercomparison Project Phase 6 (CMIP6) to further examine the Figure 9.

Turnock, S. T., Allen, R. J., Andrews, M., Bauer, S. E., Deushi, M., Emmons, L., Good, P., Horowitz, L., John, J. G., Michou, M., Nabat, P., Naik, V., Neubauer, D., O'Connor, F. M., Olivié, D., Oshima, N., Schulz, M., Sellar, A., Shim, S., Takemura, T., Tilmes, S., Tsigaridis, K., Wu, T., and Zhang, J.: Historical and future changes in air pollutants from CMIP6 models, Atmos. Chem. Phys., 20, 14547–14579,

https://doi.org/10.5194/acp-20-14547-2020, 2020.

**Response:** Thank you for your valuable comment. Our research reconstructs the annual ozone variation series during the warm season (June-September) based on weather patterns. Figure 9 shows the reconstruction results, which are primarily based on the frequency and intensity of weather patterns. Based on your suggestion, we conducted a comparative verification of our results. We find that the overall trends of the reconstructed $O_3$ sequences under future scenarios are similar to those reported in previous studies (e.g., Turnock et al., 2020, Li et al., 2023), indicating that our reconstruction reasonably reflects the future trends of $O_3$. Based on this, we revised the introduction and future reconstruction sections.

Changes in the revised manuscript (lines 102–111 and 502–516) are shown as follows:

"Affected by the surrounding atmospheric circulation, the slower the TCs move, the longer their impact duration and the greater their effects, which influence $O_3$ transport, extend the duration of $O_3$ pollution, exacerbate $O_3$ concentration, and expand the spatial extent of pollution. Global climate models (GCMs) are able to directly simulate surface $O_3$ concentrations under both historical and future scenarios (Turnock et al., 2020). These simulations provide an important reference for understanding the long-term evolution of surface $O_3$ driven by changes in emissions and climate. In the context of global warming in the future, the increase in unfavorable meteorological conditions will make the $O_3$ pollution problem more serious (Fu and Tai, 2015; Keeble et al., 2017; Arnold et al., 2018; Akritidis et al., 2019; Saunier et al., 2020). Therefore, studying the trend of $O_3$ changes under future climate change scenarios is particularly important for formulating countermeasures against $O_3$ pollution.

The average $O_3$ concentration in the YRD from June to September during 2018 to 2022 was 123.89 μg/m³. Under the SSP2-4.5 scenario, $O_3$ concentrations in the YRD are projected to increase relative to the historical period (Fig. 9a), with an average increase of approximately 1.88 μg/m³. Based on the reconstructed contribution of SWP changes, future $O_3$ concentrations in the YRD are estimated to be 2.70 μg/m³ higher than in the historical period. Under the SSP5-8.5 scenario, $O_3$ concentrations are projected to increase relative to the historical period (Fig. 9b), reaching 133.80 μg/m³

in 2100, an increase of approximately 6.86 $\mu g/m^3$. Based on the reconstructed contribution of SWP changes, future $O_3$ concentrations in the YRD are estimated to be 9.85 $\mu g/m^3$ higher than in the historical period. In summary, under both the SSP2-4.5 and SSP5-8.5 future climate scenarios, $O_3$ concentrations in the YRD are projected to increase from June to September, with more severe $O_3$ pollution under the SSP5-8.5 scenario. Previous studies based on CMIP6 multi-model simulations have shown that surface $O_3$ concentrations are projected to decrease in response to reductions in anthropogenic emissions, although the magnitude and spatial distribution of changes vary among scenarios (Turnock et al., 2020; Li et al., 2023). The trends revealed in this study are generally consistent with those of previous studies, lending confidence to the robustness of our findings."

**References**

Li, H., Yang, Y., Jin, J., Wang, H., Li, K., Wang, P., and Liao, H.: Climate-driven deterioration of future ozone pollution in Asia predicted by machine learning with multi-source data, Atmos. Chem. Phys., 23, 1131–1145, https://doi.org/10.5194/acp-23-1131-2023, 2023.

Turnock, S. T., Allen, R. J., Andrews, M., Bauer, S. E., Deushi, M., Emmons, L., Good, P., Horowitz, L., John, J. G., Michou, M., Nabat, P., Naik, V., Neubauer, D., O'Connor, F. M., Olivié, D., Oshima, N., Schulz, M., Sellar, A., Shim, S., Takemura, T., Tilmes, S., Tsigaridis, K., Wu, T., and Zhang, J.: Historical and future changes in air pollutants from CMIP6 models, Atmos. Chem. Phys., 20, 14547-14579, http://doi.org/10.5194/acp-20-14547-2020, 2020.

Finally, we would like to thank the reviewers for their comments and suggestions, which improved the rigor of our work and increased the article's novelty. Once again, thank you for your time. We hope that these revisions, made based on your comments, effectively address all of your concerns regarding our manuscript.

---

## Author Comment (AC2)

**Response to Reviewers**

*Journal:*          *Atmospheric Chemistry and Physics*

*Manuscript No.:*   egusphere-2025-2466

*Title of Paper:*   The impact of tropical cyclones on regional ozone pollution and its future trend in the Yangtze River Delta of China

*Authors:*          Mengzhu Xi, Min Xie*, Da Gao, Danyang Ma, Yi Luo, Lingyun Feng, Shitong Chen, Shuxian Zhang

Dear Editor,

We would like to submit our revised manuscript entitled "**The impact of tropical cyclones on regional ozone pollution and its future trend in the Yangtze River Delta of China**" to *Atmospheric Chemistry and Physics*.

On behalf of my co-authors, we thank you for handling the peer review of our manuscript. We appreciate the time and effort put in by you and the referees in reviewing our manuscript. These constructive comments are all valuable for revising and improving our manuscript. We have carefully studied these comments and made correction as requested. In the point-by-point responses, the reviewer's comments are in black, the author's responses are in blue, and the changes made to the text are highlighted in red. The line numbers in the author's responses are obtained from the clean version of revised manuscript, in which all the revisions have been accepted.

Best regards,

Authors

**Anonymous referee #1:**

I have carefully reviewed the manuscript entitled "The impact of tropical cyclones on regional ozone pollution and its future trend in the Yangtze River Delta of China". While the topic is of potential interest and relevance, particularly in the context of climate change and regional air quality, I cannot recommend publication in its current form. The manuscript suffers from multiple major flaws in methodology, presentation, data interpretation, and referencing, which severely undermine its scientific credibility. I therefore recommend rejection, although a substantially revised version may be reconsidered as a new submission. My major and specific concerns are listed below.

**Response:** We thank the referee #1 for the constructive comments and suggestions, which are very helpful for improving the clarity and reliability of the manuscript. We modified the methods, reanalyzed the data, rewrite the sentences for better readability, and enhanced the clarity of figures, to eliminate the mentioned major flaws in methodology, presentation, data interpretation, and referencing. Our point-by-point responses to the specific comments are listed as below.

**Major concerns:**

1. The authors identify five synoptic weather patterns (SWPs) using a rotated principal component analysis (PTT method), but they do not provide adequate information on the physical characteristics of these SWPs or justify the use of different meteorological parameters and domains in defining their intensities (Lines 167–171, 259–264).

**Response:** We thank the reviewer for the valuable comment. In the revised manuscript, we have made the following clarifications and improvements.

1. On the physical characteristics of the SWPs (Lines 259–264):

In the original manuscript, to focus on the main topic, only the SWP related to typhoon were analyzed in detail. The physical characteristics of the SWPs were briefly described, but the presentation was not sufficiently clear. Considering the comment of anonymous referee #1, in the revised manuscript, we have provided a clearer description of the typical characteristics of each classified SWP in Section 3.2, including the associated meteorological factors (such as air temperature, air humidity,

and wind speed) and the $O_3$ concentration characteristics under different SWPs. In addition, the atmospheric circulation structure at 1000 hPa, 850 hPa and 500 hPa are presented and analyzed in Figures 3 and 4, which further reveal the dynamical and thermodynamical characteristics of each SWP. These analyses together improve and complement the characterization of the physical features of the identified SWPs.

Changes in the revised manuscript (lines 260–283) are shown as follows:

3.2 Main synoptic weather patterns in the YRD

"Based on the analysis of $O_3$ variation characteristics in the YRD, $O_3$ concentrations were generally high from June to September, coinciding with the peak season of TC activity. We selected June to September during 2018−2022 as the research period and used the PTT weather classification method to classify the weather situation. SWPs in the YRD during this period were primarily divided into four categories. As shown in Table 1, SWP1 is the main SWP, occurring on 383 days and accounting for 62.79 % of the period. SWP2 and SWP4 followed, occurring on 81 and 80 days, respectively, with frequencies of 13.28 % and 13.11 %. SWP3 was less frequent, occurring on 10.82 % of days. Specifically, SWP1 was mainly influenced by the southwesterly flow introduced by the WPSH and the northeast China low, SWP2 by northwesterly flow introduced by a continental high and the northeast China low, SWP3 by southeasterly flow introduced by the WPSH, and SWP4 by northeasterly flow introduced by the WPSH and TCs.

Statistics of average $O_3$ concentrations and meteorological factors under each SWP indicate that SWP4 had the lowest average $O_3$ concentration (115.07 μg/m$^3$), whereas SWP2 had the highest value (132.36 μg/m$^3$). $O_3$ concentrations during SWP1 and SWP3 were similar, at 123.81 μg/m$^3$ and 123.28 μg/m$^3$, respectively. However, significant differences in $O_3$ concentrations were observed among SWPs during the same period, highlighting the strong influence of circulation patterns on $O_3$ pollution levels. SWP1 exhibited higher temperatures (27.12 °C), favoring increased $O_3$ concentrations. SWP2 was influenced by dry northwesterly airflow, exhibiting lower humidity (76.74 %) and slower wind speed (2.03 m/s), which hindered air pollutant dispersion and resulted in higher $O_3$ concentrations. SWP3 was characterized by lower

humidity (78.53 %) and slower wind speed (2.11 m/s), which were key factors contributing to increased $O_3$ concentration. Under SWP4, though TCs far from the coastline could lead to regional $O_3$ pollution (in section 3.3.2), the landfalling TCs could bring strong winds and rainstorms, increasing humidity (80.03 %) and wind speed (2.64 m/s), which resulted in lower average $O_3$ concentrations."

2. On the justification of the selected intensity factors (Lines 167–171):

We sincerely apologize that the original manuscript did not adequately explain the rationale for the choice of different meteorological parameters and spatial domains in defining SWP intensities. In previous studies, a single intensity factor was often used for multiple weather patterns; however, this sometimes led to weak correlations with the ozone interannual variation series of a specific pattern, which in turn significantly affected the subsequent reconstruction. In our revised analysis, we took into account the unique features of each weather pattern when defining its intensity factor, and we further verified the rationality of the selection by calculating the correlation coefficients between candidate intensity factors and the ozone interannual variation series under each pattern. In the revised manuscript, detailed explanations have been added in Section 3.4.1, thereby providing a stronger basis for the construction of SWP intensity indices.

Changes in the revised manuscript (lines 427–444) are shown as follows:

3.4.1 The role of changes in the intensity and frequency of SWPs in the reconstruction of the annual variation series of $O_3$

"In reconstructing the time series of annual $O_3$ concentrations, we found that the definition of the SWP intensity factor played a crucial role in the reconstruction. Previous studies reconstructed the annual $O_3$ concentration series by defining the SWP intensity factor as either the regional mean sea level pressure or the regional minimum pressure (Hegarty et al., 2007; Liu et al., 2019). However, this definition showed poor correlation between the SWP intensity factor and the annual $O_3$ concentration series for certain SWPs. Therefore, we defined the SWP intensity factor for each SWP based on its specific meteorological characteristics, selecting the maximum, minimum, and mean geopotential heights across nine zones, and evaluated its validity by calculating the

correlation coefficient with the annual $O_3$ variation series (Table 3). For SWP1 and SWP2, the maximum geopotential heights in zone 7 and zone 2 were highly correlated with the annual $O_3$ variation series. For SWP3 and SWP4, the minimum geopotential heights in zone 9 and zone 4 were highly correlated with the annual $O_3$ variation series. The maximum geopotential height reflects regional wind speeds, which determine the amount of water vapor transported into the region. Compared with SWP1, SWP2 has a larger weather system scale, so the maximum geopotential height in zone 2 shows a stronger correlation with the $O_3$ series than that in zone 7. For SWP4, the YRD was affected by TCs, and $O_3$ concentrations were closely related to TC intensity. The minimum geopotential height in zone 4 reflects the TC intensity. When the SWP intensity factor was defined based on the unique meteorological characteristics of each SWP, the reconstructed series more accurately reflected the impact of changes in SWP intensity on $O_3$ concentrations."

**Table 3.** Correlation coefficients between the $O_3$ concentration time series and different SWP intensity factors under each SWP.

| Type | Zone 1 (115°~135°E, 20°~40°N) | | | Zone 2 (90°~140°E, 20°~50°N) | | | Zone 3 (110°~130°E, 10°~40°N) | | |
|------|------|------|------|------|------|------|------|------|------|
|      | Mean | Max | Min | Mean | Max | Min | Mean | Max | Min |
| SWP1 | -0.62 | -0.72 | -0.41 | -0.69 | 0.26 | 0.45 | -0.59 | -0.76 | 0.17 |
| SWP2 | -0.65 | -0.87 | -0.45 | -0.47 | **-0.88** | -0.09 | -0.60 | -0.84 | -0.53 |
| SWP3 | -0.41 | -0.67 | 0.72 | 0.10 | -0.17 | 0.43 | -0.11 | -0.60 | 0.75 |
| SWP4 | 0.15 | -0.32 | 0.36 | 0.26 | 0.01 | 0.32 | 0.33 | -0.38 | 0.49 |

| Type | Zone 4 (110°~130°E, 25°~40°N) | | | Zone 5 (100°~120°E, 15°~35°N) | | | Zone 6 (110°~120°E, 15°~35°N) | | |
|------|------|------|------|------|------|------|------|------|------|
|      | Mean | Max | Min | Mean | Max | Min | Mean | Max | Min |
| SWP1 | -0.67 | -0.80 | -0.02 | -0.33 | 0.37 | 0.05 | -0.60 | -0.82 | 0.10 |
| SWP2 | -0.47 | -0.73 | -0.44 | -0.36 | -0.59 | -0.19 | -0.48 | -0.65 | -0.29 |
| SWP3 | -0.19 | -0.60 | 0.46 | 0.34 | 0.31 | 0.67 | 0.29 | -0.29 | 0.85 |
| SWP4 | 0.31 | -0.32 | **0.64** | 0.48 | -0.02 | 0.55 | 0.47 | 0.26 | 0.55 |

| Type | Zone 7 (110°~130°E, 20°~35°N) | | | Zone 8 (115°~135°E, 30°~50°N) | | | Zone 9 (110°~130°E, 20°~40°N) | | |
|------|------|------|------|------|------|------|------|------|------|
|      | Mean | Max | Min | Mean | Max | Min | Mean | Max | Min |
| SWP1 | -0.53 | **-0.83** | 0.06 | -0.40 | -0.73 | 0.42 | -0.61 | -0.76 | 0.09 |
| SWP2 | -0.54 | -0.79 | -0.25 | -0.31 | -0.72 | 0.03 | -0.56 | -0.77 | -0.50 |
| SWP3 | 0.02 | -0.69 | 0.86 | -0.25 | -0.64 | 0.33 | -0.08 | -0.60 | **0.87** |

| | | | | | | | | | |
|---|---|---|---|---|---|---|---|---|---|
| SWP4 | 0.50 | -0.21 | 0.49 | -0.19 | -0.03 | -0.15 | 0.39 | -0.30 | 0.48 |

Furthermore, Table 1 suggests that some SWPs primarily occur in winter, spring, and autumn, while SWP2 and SWP5 dominate in summer. This implies a strong seasonal signal embedded in the classification. However, the authors proceed to analyses $O_3$ pollution impacts without removing seasonal cycles, which is inappropriate given the strong seasonal variability in both $O_3$ precursors and photochemistry. Weather pattern classification should be conducted on seasonally detrended data to avoid conflating synoptic and seasonal influences.

**Response:** Thank you for your meticulous review and valuable comments on our manuscript. We understand the concern regarding the potential influence of seasonal signals on the analysis of SWPs and $O_3$. In the revised manuscript, we have taken this into consideration and provided a clear explanation. For a year-round analysis, to perform seasonally detrended analysis is indeed reasonable. However, the focus of this study is on the impact of tropical cyclones on $O_3$ pollution. Our analysis shows that $O_3$ pollution in the Yangtze River Delta predominantly occurs from June to September, which coincides with the peak period of tropical cyclone activity. Therefore, we have adjusted the study period to **June–September** and conducted weather pattern classification only for this period, excluding year-round data. This adjustment effectively minimizes the influence of seasonal variability on the analysis results. It also ensures that the classification is focused on the relevant conditions for assessing the impact of tropical cyclones on $O_3$, reducing the confounding effects of seasonal changes on $O_3$ precursors and photochemistry, and providing a more robust basis for evaluating the role of weather patterns in $O_3$ pollution.

Changes in the revised manuscript (lines 260–270) are shown as follows:

3.2 Main synoptic weather patterns in the YRD

"Based on the analysis of $O_3$ variation characteristics in the YRD, $O_3$ concentrations were generally high from June to September, coinciding with the peak season of TC activity. We selected June to September during 2018−2022 as the research period and used the PTT weather classification method to classify the weather situation.

SWPs in the YRD during this period were primarily divided into four categories. As shown in Table 1, SWP1 is the main SWP, occurring on 383 days and accounting for 62.79 % of the period. SWP2 and SWP4 followed, occurring on 81 and 80 days, respectively, with frequencies of 13.28 % and 13.11 %. SWP3 was less frequent, occurring on 10.82 % of days. Specifically, SWP1 was mainly influenced by the southwesterly flow introduced by the WPSH and the northeast China low, SWP2 by northwesterly flow introduced by a continental high and the northeast China low, SWP3 by southeasterly flow introduced by the WPSH, and SWP4 by northeasterly flow introduced by the WPSH and TCs."

2. Many key numerical conclusions lack both graphical support and proper referencing (e.g., Lines 394–404, 436–438). For instance, the stated contributions of SWP5 to $O_3$ concentration changes—15.14% in 2030 and 20.66% in 2060—are not clearly contextualized. These values appear to indicate that SWP5 is the most influential within those respective years compared to other years, rather than compared to other weather patterns. However, the manuscript does not clearly define the comparison baseline, which may mislead readers into interpreting these percentages as reflecting the dominance of SWP5 over other synoptic types. Clarification is essential to avoid misinterpretation.

**Response:** We thank the reviewer for this careful comment. We sincerely apologize that some numerical conclusions in the original manuscript lacked graphical support and were not clearly presented, which could potentially lead to misinterpretation. In the revised manuscript, we have clarified and improved these points as follows.

1. To address the issue of key numerical conclusions lacking graphical support, we have updated the relevant figures in the revised manuscript to provide visual support for these results (e.g., Figure 7). These updates allow readers to more clearly understand the numerical values and enhance the visualization and interpretability of the conclusions. Changes in the revised manuscript (lines 463–472) are shown as follows:

3.4.2 Changes in TCs and the effects on $O_3$ concentration over the YRD in the future

"Under the SSP2-4.5 scenario, the frequencies of SWP1 to SWP4 were 57.79%, 19.67%, 6.76%, and 15.78%, respectively (Fig. 10). High-average $O_3$ pattern occurred on 93, 99, 96, and 90 days in 2030, 2035, 2060, and 2100, respectively, showing a general increasing trend, with the peak occurring in 2035. These results suggest that, considering only changes in SWP frequency, $O_3$ concentrations would reach a maximum in 2035 under the SSP2-4.5 scenario. Under the SSP5-8.5 scenario, the frequencies of SWP1 to SWP4 were 50.82%, 23.98%, 7.17%, and 18.03%, respectively (Fig. 10). High-average $O_3$ pattern will occur on 99, 90, 82, and 94 days in 2030, 2035, 2060, and 2100, respectively, under the SSP5-8.5 scenario, with the number of days increasing in 2030 and 2100. This also suggests that, considering only changes in SWP frequency, $O_3$ concentrations would increase in 2030 and 2100 under the SSP5-8.5 scenario."

[Figure]

**Figure 7.** Distribution of the number of days of occurrence of each SWP in the YRD from June to September under the historical period (2018-2022), SSP2-4.5, and SSP5-8.5 future scenarios.

2. The reported percentages (e.g., 15.14% in 2030 and 20.66% in 2060) represent the relative contributions of each SWP to $O_3$ changes within the same year, rather than direct comparisons across different weather patterns. We have revised the wording in the manuscript to make it clear that historical tropical cyclone contributions serve as

the comparison baseline, thereby preventing potential misinterpretation. To further enhance the visualization and interpretability, we have added the relevant figure (Figure 10b) and included necessary references in the text, allowing readers to intuitively understand the contributions of each weather pattern to $O_3$ changes. These revisions ensure that the conclusions are clearly presented and readily understandable.

Changes in the revised manuscript (lines 521–535) are shown as follows:

3.4.3 Reconstruction of annual $O_3$ variability in the YRD under future scenarios

"Figure 10a shows the contribution of each SWP to the reconstructed series of future $O_3$ concentrations in the YRD from June to September under the SSP2-4.5 and SSP5-8.5 scenarios. We focus on the contribution of the TC weather pattern (SWP4) to the reconstructed $O_3$ concentration series under different scenarios (Figure 10b). During the historical period (2018−2022), the TC weather pattern contributed an average of 13.11% to $O_3$ concentrations in the YRD. Under the SSP2-4.5 scenario, the contributions of the TC weather pattern in 2030, 2035, 2060, and 2100 were 15.57%, 13.93%, 17.21%, and 16.39%, respectively. The frequency of the TC weather pattern in the YRD in 2035 was lower, resulting in a lower contribution of the TC weather pattern to $O_3$ concentrations in that year. Under the SSP5-8.5 scenario, the TC weather pattern contributes 14.75%, 15.57%, 24.59%, and 19.67% to $O_3$ changes in 2030, 2035, 2060, and 2100, respectively. Under both the SSP2-4.5 and SSP5-8.5 scenarios, the contribution of the TC weather pattern to $O_3$ pollution increases to varying degrees compared with the historical period. This suggests that under future climate change, the impact of TC weather pattern on $O_3$ pollution in the YRD may intensify. Further research is needed on the relationship between key SWPs, such as TC weather pattern, and $O_3$ formation mechanisms to more accurately predict and mitigate regional $O_3$ pollution under future climate conditions."

[Figure]

**Figure 10.** Contribution of SWP to the projected annual variation series of O$_3$ concentration under the SSP2-4.5 and SSP5-8.5 future scenarios. **(a)** Contribution of four SWPs, **(b)** Contribution of TC weather pattern (SWP4).

Several critical results, such as the contribution index and the role of TC-generated meteorological conditions (e.g., "low humidity and strong solar radiation"), lack quantitative evidence (Lines 309–311). This weakens the entire discussion on the role of TCs in driving O$_3$ changes.

**Response:** We thank the reviewer for this insightful comment. Regarding the problem that "the contribution index and the role of TC-generated meteorological conditions lack quantitative evidence", we have provided clarifications and additional details in the revised manuscript.

1. We recognize that the original manuscript on lines 309–311 may have caused misunderstanding, implying that "tropical cyclones directly cause low humidity and strong solar radiation". In fact, our intention was to describe the summer meteorological conditions themselves—characterized by high temperature, low humidity, and strong radiation—which provide favorable conditions for O$_3$ formation, rather than being directly induced by tropical cyclones. To avoid such misinterpretation, we have revised the relevant text in the revised manuscript and analyzed the physical characteristics of each weather pattern during the selected study period, clearly distinguishing the effects of tropical cyclone–induced changes on O$_3$.

Changes in the revised manuscript (lines 318–328) are shown as follows:

3.3.1 The impact of TC weather pattern (SWP4) on $O_3$ pollution in the YRD

"Figure 4 illustrates the three-dimensional atmospheric circulation structure under the TC weather pattern (SWP4). For the TC weather pattern, similar circulation conditions were observed at 850 hPa (Fig. 4a) and at sea level (Fig. 4b). The YRD was located northwest of the TCs and was controlled by northeasterly flow guided by the TCs. The direction and intensity of the northeast wind had a significant impact on meteorological conditions and pollutant transport in the YRD. At 500 hPa, the region was dominated by westerly or northwesterly flow (Fig. 4c). Meanwhile, the peripheral downward airflow associated with lower-level TCs (Fig. 4d) led to a more stagnant atmosphere over the YRD. As the TC approaches the YRD, strong northeasterly flow increased clean sea airflow transportation to the region, lowered temperatures and increased humidity, creating unfavorable meteorological conditions for photochemical reactions. Furthermore, higher wind speeds facilitated air pollutant elimination, leading to a decrease in $O_3$ concentrations in the region (Table 1)."

2. The description of the contribution index in the original manuscript was not sufficiently clear, leading to ambiguity in its mathematical definition and physical interpretation. To accurately quantify the impacts of changes in SWP frequency and intensity on interannual $O_3$ variations, we calculated the contribution index as the ratio of the interannual amplitude of the reconstructed sequence (maximum minus minimum $O_3$ concentration within the reconstructed sequence) to that of the original sequence (maximum minus minimum $O_3$ concentration in the original series). This index represents the proportion of interannual $O_3$ variation explained by a given factor (e.g., changes in SWP frequency or combined frequency and intensity). The results indicate that when considering only SWP frequency changes, the reconstructed contribution is 10.05%, whereas including SWP intensity changes increases the reconstructed contribution to 69.66%, demonstrating that intensity variations have a much stronger influence on interannual $O_3$ differences than frequency variations. We have revised the manuscript accordingly to make the definition, calculation, and physical meaning of the contribution index clear to readers.

Changes in the revised manuscript (lines 408–421) are shown as follows:

3.4.1 The role of changes in the intensity and frequency of SWPs in the reconstruction of the annual variation series of $O_3$

"When only changes in SWP frequency were considered, the reconstructed time series was relatively flat and did not adequately capture the variation trend of $O_3$ concentrations. Therefore, changes in SWP frequency had minimal impact on the annual variation of $O_3$. When changes in SWP intensity were considered, the reconstructed series more closely resembled the original annual variation series. Therefore, compared to changes in SWP frequency, changes in SWP intensity contributed more to variations in $O_3$ concentration. To accurately assess the impact of both SWP frequency and intensity on annual $O_3$ variation, we quantitatively calculated their contributions. The contribution index was defined as the ratio of the interannual variation amplitude of the reconstructed series to that of the original series, i.e., ($O_{3max}$ of the reconstructed series $-$ $O_{3min}$ of the reconstructed series) / ($O_{3max}$ of the original series $-$ $O_{3min}$ of the original series). When only changes in SWP frequency were considered, their contribution to the interannual variation was 10.05%. When changes in SWP intensity were additionally included, the contribution increased to 69.66%. This indicates that, compared with changes in SWP frequency, changes in SWP intensity played a more important role in driving interannual variations in $O_3$ concentrations."

3. Throughout the manuscript—particularly in the introduction—numerous claims are either incorrectly cited, misrepresented, or not supported by the referenced sources. This significantly undermines the scientific credibility and rigour of the work.

**Response:** Thank you very much for your careful review of our research and your valuable feedback. We are deeply sorry for the issues you pointed out, including inappropriate citations, misinterpretations, and lack of supporting references. We attach great importance to this and have carefully reviewed and analyzed the manuscript. During this process, we found that some of the issues may be due to unclear presentation or misunderstandings. To address these misunderstandings, we have reorganized and revised the relevant paragraphs, striving for greater clarity and logic. We believe that these revisions better convey our views and research ideas while

enhancing the scientific rigor of our research.

Here are some examples, lines 38–40: The statement that $O_3$ accounted for over 50% of polluted days is not substantiated by the referenced China Ecological Environment Bulletin. The authors should provide the correct citation, improve the clarity of the English, and revise the conclusion to reflect the actual content of the report.

**Response:** Thank you for pointing this out. After checking, we confirm that the phrase "in many Chinese cities" in the original manuscript was inaccurate. The data we cited are from the 2020 China Ecological Environment Bulletin for the Yangtze River Delta region, where $O_3$ was the primary pollutant on 50.7% of the total number of polluted days. We have modified the sentence in the revised manuscript.

Changes in the revised manuscript (lines 47–49) are shown as follows:

"…According to the 2020 China Ecological Environment Bulletin, in the YRD, the number of days with air quality exceeding the standard with $O_3$ as the primary pollutant accounted for over 50% of the total number of days exceeding the standard."

Lines 77–80: Assertions regarding the low-frequency modulation of tropical cyclones (TCs) by climate change are made without any credible or peer-reviewed references.

**Response:** Thank you for your valuable comment. We sincerely apologize for not providing relevant references earlier. Thank you for your valuable comment. We have now included authoritative peer-reviewed literature to strengthen the scientific basis of our discussion regarding the low-frequency modulation of tropical cyclones by climate change.

Changes in the revised manuscript (lines 86–90) are shown as follows:

"…In the context of global climate change, with the increases in greenhouse gas emissions, global warming, and interannual climate changes, the state of the atmosphere and ocean has changed, and sea-level rise and extreme climate events have occurred frequently, which have a low-frequency modulating effect on TCs (Chu et al., 2020; Moon et al., 2023; Moon et al., 2025; Wang et al., 2023a) …"

**References**

Chu, J. E., Lee, S. S., Timmermann, A., Wengel, C., Stuecker, M. F., and Yamaguchi,

R.: Reduced tropical cyclone densities and ocean effects due to anthropogenic greenhouse warming, Science Advances, 6, http://doi.org/10.1126/sciadv.abd5109, 2020.

Moon, M., Ha, K. J., Kim, D., Ho, C. H., Park, D. S. R., Chu, J. E., Lee, S. S., and Chan, J. C. L.: Rainfall strength and area from landfalling tropical cyclones over the North Indian and western North Pacific oceans under increased CO2 conditions, Weather and Climate Extremes, 41, http://doi.org/10.1016/j.wace.2023.100581, 2023.

Moon, M., Min, S. K., Chu, J. E., An, S. I., Son, S. W., Ramsay, H., and Wang, Z.: Tropical cyclone response to ambitious decarbonization scenarios, Npj Climate and Atmospheric Science, 8, http://doi.org/10.1038/s41612-025-01122-9, 2025.

Wang, S., Murakami, H., and Cooke, W. F.: Anthropogenic forcing changes coastal tropical cyclone frequency, Npj Climate and Atmospheric Science, 6, http://doi.org/10.1038/s41612-023-00516-x, 2023a.

Lines 86–88: The claim that TC frequency in the Northwest Pacific has decreased, while intensity and duration have increased, is not convincingly supported by the cited literature. In fact, some references appear to be news media or secondary summaries rather than original peer-reviewed scientific studies.

**Response:** Thank you very much for your careful review of our manuscript and your valuable comments. Regarding the references on changes in the frequency, intensity, and duration of tropical cyclones in the Northwest Pacific, we have carefully re-examined the relevant literature to avoid any possible misunderstanding. We would like to clarify that all the references cited in the manuscript are peer-reviewed scientific articles rather than news media or secondary summaries. In the original manuscript, Yu et al. (2025) was cited to support the view that "tropical cyclones in the western Pacific are becoming more intense". However, this article does not directly examine changes in tropical cyclone intensity. Instead, its introduction provides a broader context for the increasing intensity of tropical cyclones in the western Pacific. Therefore, we have replaced it with a more appropriate reference. Recent peer-reviewed studies (e.g.,

(Balaguru et al., 2024; Bhatia et al., 2022; Chand et al., 2022; Jung, 2025; Wang et al., 2022a; Kossin, 2018; Zhang et al., 2020) indicate that under global warming, the frequency of tropical cyclones in the western Pacific is decreasing, but their average intensity is increasing, and their duration is also increasing.

Changes in the revised manuscript (lines 96–99) are shown as follows:

"…Although the frequency of TCs in the northwest Pacific Ocean has decreased in recent years, their average intensity has shown an upward trend (Balaguru et al., 2024; Bhatia et al., 2022; Chand et al., 2022; Jung, 2025; Wang et al., 2022a) and their duration has also become longer (Kossin, 2018; Zhang et al., 2020)…"

**References**

Balaguru, K., Chang, C. C., Leung, L. R., Foltz, G. R., Hagos, S. M., Wehner, M. F., Kossin, J. P., Ting, M. F., and Xu, W. W.: A Global Increase in Nearshore Tropical Cyclone Intensification, Earths Future, 12, http://doi.org/10.1029/2023ef004230, 2024.

Bhatia, K., Baker, A., Yang, W. C., Vecchi, G., Knutson, T., Murakami, H., Kossin, J., Hodges, K., Dixon, K., Bronselaer, B., and Whitlock, C.: A potential explanation for the global increase in tropical cyclone rapid intensification, Nature Communications, 13, http://doi.org/10.1038/s41467-022-34321-6, 2022.

Chand, S. S., Walsh, K. J. E., Camargo, S. J., Kossin, J. P., Tory, K. J., Wehner, M. F., Chan, J. C. L., Klotzbach, P. J., Dowdy, A. J., Bell, S. S., Ramsay, H. A., and Murakami, H.: Declining tropical cyclone frequency under global warming, Nature Climate Change, 12, 655-+, http://doi.org/10.1038/s41558-022-01388-4, 2022.

Jung, H.: Humans fuel stronger cyclones, Nature Climate Change, 15, 351-351, http://doi.org/10.1038/s41558-025-02321-1, 2025.

Kossin, J. P.: A global slowdown of tropical-cyclone translation speed, Nature, 558, 104-+, http://doi.org/10.1038/s41586-018-0158-3, 2018.

Wang, G. H., Wu, L. W., Mei, W., and Xie, S. P.: Ocean currents show global intensification of weak tropical cyclones, Nature, 611, 496-+, http://doi.org/10.1038/s41586-022-05326-4, 2022a.

Zhang, G., Murakami, H., Knutson, T. R., Mizuta, R., and Yoshida, K.: Tropical cyclone motion in a changing climate, Science Advances, 6, http://doi.org/10.1126/sciadv.aaz7610, 2020.

**Specific Comments**

1. Line 50: Please spell out "TC" at first mention.

**Response:** Thank you for pointing this out. We have spelled out "TC" at its first occurrence as "tropical cyclone (TC)" and retained the abbreviation consistently thereafter.

Changes in the revised manuscript (lines 59–60) are shown as follows:

"Tropical cyclone (TC) activities have a profound impact on the ecological environment of China's coastal areas. …"

2. Lines 190–192: Claims on $O_3$ variation lack data or plots. Add evidence.

**Response:** Thank you for your careful review of our manuscript and your scientific suggestions. According to your suggestions, we added this part to the revised manuscript, enhancing the rationality and completeness of our analysis. We have also modified and optimized Figure 1, adding trend lines to more clearly illustrate the trend of $O_3$ concentration. Key data points, such as the peaks and valleys of $O_3$ concentration, are detailed in the figure, with the corresponding concentration values highlighted at peak locations for easier understanding. Furthermore, based on the original figure, we have added temporal trends in meteorological factors such as solar radiation, temperature, precipitation, and relative humidity to further support our assertion that meteorological conditions drive $O_3$ changes.

Changes in the revised manuscript (lines 229–235) are shown as follows:

[Figure]

**Figure 1.** Inter-monthly variations in mean $O_3$ concentrations and associated meteorological conditions in the YRD from 2018 to 2022. **(a)** The monthly variation of $O_3$ concentrations is shown, with the red trend line representing the monthly mean values and the individual monthly concentrations labeled. **(b-c)** The variation trends of solar radiation, temperature, precipitation and relative humidity during the same period. (The shaded area in the violin plot represents the kernel density curve, with the three black lines indicating the 25th percentile, the mean, and the 75th percentile, respectively.)

3. Lines 213–217: The description of two TC-$O_3$ response types is unsupported. Provide composite analysis and classify examples accordingly.

**Response:** We thank the reviewer for this valuable comment. Following your suggestion, we have clarified the relevant section in the revised manuscript. In particular, Figure 2 provides a clear visualization of the $O_3$ concentration changes during tropical cyclone events, where two typical patterns can be observed: one showing an increase–decrease sequence, and the other showing an increase–decrease–increase sequence. These trends indicate that tropical cyclones indeed influence $O_3$ concentrations, providing a theoretical basis for our subsequent analyses. The main purpose of this section was to highlight the impact of tropical cyclones on $O_3$, rather than to conduct an in-depth exploration of specific "TC-$O_3$ response types". Therefore, we have simplified and clarified the wording in the revised manuscript to avoid potential ambiguity.

Changes in the revised manuscript (lines 236–254) are shown as follows:

3.1 Characteristics of $O_3$ pollution in the YRD

"$O_3$ pollution in the YRD is concentrated in warm months of the year. We analyzed the temporal distribution of $O_3$ concentrations in representative cities in the YRD (Shanghai, Nanjing, Hangzhou, and Hefei) from April to September during 2018–2022 (Fig. 2). Since TCs mainly occur after June, $O_3$ pollution is often associated with TC activity. During the same TC period, the $O_3$ concentrations in Shanghai, Nanjing, Hangzhou, and Hefei were different, but the temporal trends were similar. TCs have a certain impact on the changes in $O_3$ concentrations in the YRD. According to the evolution and trajectory of TC weather, TCs affecting China are primarily generated in the northwest Pacific Ocean (Zhan et al., 2020; Wang et al., 2024a). As they develop and move, these TCs will have a significant impact on $O_3$ concentrations in China. At this time, the YRD is located on the periphery of the TCs. Under the control of the periphery of the TCs, the strong downward airflow will make the YRD in stagnant weather conditions and inhibit the diffusion of pollutants (Shu et al., 2016), accompanied by high temperature, clear and dry weather conditions, which are the main weather conditions causing $O_3$ pollution. With the evolution of TC weather system, when the TCs center gradually approaches the YRD until it is within a certain range, the YRD is no longer under the influence of the periphery of the TCs and comes under the control of the TC wind and rain belt. Strong winds and precipitation can significantly cleanse air pollutants, and thereby reduce $O_3$ concentrations. After TCs dissipate, $O_3$ pollution levels may rise again due to restored meteorological conditions conducive to $O_3$ formation (Zhan et al., 2020). These results suggest that TCs can significantly affect $O_3$ pollution in the YRD, highlighting the importance of implementing $O_3$ control measures prior to TC landfall."

4. Lines 217–218: No references for TC genesis location or influence radius. Clarify language: "generation" implies early-stage oceanic development, likely too far from YRD to affect local $O_3$.

**Response:** We thank the reviewer for this helpful comment. We acknowledge that the original manuscript in Lines 217–218 may cause ambiguity. Our intention was to state

that the tropical cyclones affecting the YRD mainly originate in the northwest Pacific and subsequently influence the regional atmosphere through their development and trajectories, rather than implying that their early-stage oceanic genesis directly affects local $O_3$. To avoid confusion, we have revised the wording in the manuscript and added appropriate references (e.g., Zhan et al., 2020; Wang et al., 2024) to support the description of the typical genesis locations and influence range of TCs.

Changes in the revised manuscript (lines 241–248) are shown as follows:

3.1 Characteristics of $O_3$ pollution in the YRD

"…According to the evolution and trajectory of TC weather, TCs affecting China are primarily from the northwest Pacific Ocean (Zhan et al., 2020; Wang et al., 2024a). As they develop and move, these TCs will have a significant impact on $O_3$ concentrations in China (Xi et al., 2025). At this time, the YRD is located on the periphery of the TCs. Under the control of the periphery of the TCs, the strong downward airflow will make the YRD in stagnant weather conditions and inhibit the diffusion of pollutants (Shu et al., 2016), accompanied by high temperature, clear and dry weather conditions, which are the main weather conditions causing $O_3$ pollution …"

**References**

Wang, J. H., Wang, P., Tian, C. F., Gao, M., Cheng, T. T., and Mei, W.: Consecutive Northward Super Typhoons Induced Extreme Ozone Pollution Events in Eastern China, Npj Climate and Atmospheric Science, 7, http://doi.org/10.1038/s41612-024-00786-z, 2024a.

Xi, M., Luo, Y., Li, Y., Ma, D., Feng, L., Zhang, S., Chen, S., and Xie, M.: Comprehensive analysis of prevailing weather patterns and high-impact typhoon tracks to reveal where and how tropical cyclone affects regional ozone pollution in the Yangtze River Delta region, China, Atmospheric Environment, 361, 121498, http://doi.org/https://doi.org/10.1016/j.atmosenv.2025.121498, 2025.

Zhan, C. C., Xie, M., Huang, C. W., Liu, J. N., Wang, T. J., Xu, M., Ma, C. Q., Yu, J. W., Jiao, Y. M., Li, M. M., Li, S., Zhuang, B. L., Zhao, M., and Nie, D. Y.: Ozone affected by a succession of four landfall typhoons in the Yangtze River Delta, China: major processes and health impacts, Atmospheric Chemistry and Physics,

20, 13781-13799, http://doi.org/10.5194/acp-20-13781-2020, 2020.

5. Lines 227–229: Claims not supported by figures. Add case studies or remove.

**Response:** Thank you for the reviewer's comments. Although these claims are supported by figures and data in a later section of our manuscript, it is not directly supported by the data or figures presented in the context of Lines 227–229. To maintain logical rigor and avoid potential misunderstanding at this stage, we have removed the claims from this section.

6. Lines 309–311: The link between TC conditions and "low humidity, strong solar radiation" is unsubstantiated. TCs often increase regional moisture. Provide observational support.

**Response:** Thank you for the reviewer's insightful comments on our manuscript. We recognize that the original manuscript in Lines 309–311 may have caused misunderstanding, implying that "tropical cyclones directly cause low humidity and strong solar radiation". In fact, our intention was to describe the summer meteorological conditions themselves—characterized by high temperature, low humidity, and strong radiation—which provide favorable conditions for $O_3$ formation, rather than being directly induced by tropical cyclones. To avoid such misinterpretation, we have revised the relevant text in the manuscript and analyzed the physical characteristics of each weather pattern during the selected study period, clearly distinguishing the effects of tropical cyclone–induced changes on $O_3$.

Changes in the revised manuscript (lines 318–328) are shown as follows:

3.3.1 The impact of TC weather pattern (SWP4) on $O_3$ pollution in the YRD

"Figure 4 illustrates the three-dimensional atmospheric circulation structure under the TC weather pattern (SWP4). For the TC weather pattern, similar circulation conditions were observed at 850 hPa (Fig. 4a) and at sea level (Fig. 4b). The YRD was located northwest of the TCs and was controlled by northeasterly flow guided by the TCs. The direction and intensity of the northeast wind had a significant impact on meteorological conditions and pollutant transport in the YRD. At 500 hPa, the region

was dominated by westerly or northwesterly flow (Fig. 4c). Meanwhile, the peripheral downward airflow associated with lower-level TCs (Fig. 4d) led to a more stagnant atmosphere over the YRD. As the TC approaches the YRD, strong northeasterly flow increased clean sea airflow transportation to the region, lowered temperatures and increased humidity, creating unfavorable meteorological conditions for photochemical reactions. Furthermore, higher wind speeds facilitated air pollutant elimination, leading to a decrease in $O_3$ concentrations in the region (Table 1)."

7. Line 348: The 500 hPa ridge discussion (Fig. 5g–i) is unclear. Differences between panels are not visually apparent. Consider replotting with clearer contrasts.

**Response:** Thank you for pointing this out. In response to the concern that the discussion of the 500 hPa ridge line (Fig. 5g–i) was unclear and that the differences were difficult to distinguish, we have redefined the study periods and redrawn the corresponding figures in the revised manuscript. These modifications highlight the contrasts more effectively and make the results clearer and easier to interpret.

8. Lines 368–372: The "contribution index" needs a mathematical definition and justification. The interpretation lacks physical meaning without context.

**Response:** Thank you for your valuable comment. The description of the contribution index in the original manuscript was not sufficiently clear, leading to ambiguity in its mathematical definition and physical interpretation. In this study, to accurately quantify the impacts of changes in SWP frequency and intensity on interannual $O_3$ variations, we calculated the contribution index as the ratio of the interannual amplitude of the reconstructed sequence (maximum minus minimum $O_3$ concentration within the reconstructed sequence) to that of the original sequence (maximum minus minimum $O_3$ concentration in the original series). This index represents the proportion of interannual $O_3$ variation explained by a given factor (e.g., changes in SWP frequency or combined frequency and intensity). The results indicate that when considering only SWP frequency changes, the reconstructed contribution is 10.05%, whereas including SWP intensity changes increases the reconstructed contribution to 69.66%, demonstrating

that intensity variations have a much stronger influence on interannual $O_3$ differences than frequency variations. We have revised the manuscript accordingly to make the definition, calculation, and physical meaning of the contribution index clear to readers. Changes in the revised manuscript (lines 408–421) are shown as follows:

3.4.1 The role of changes in the intensity and frequency of SWPs in the reconstruction of the annual variation series of $O_3$

"When only changes in SWP frequency were considered, the reconstructed time series was relatively flat and did not adequately capture the variation trend of $O_3$ concentrations. Therefore, changes in SWP frequency had minimal impact on the annual variation of $O_3$. When changes in SWP intensity were considered, the reconstructed series more closely resembled the original annual variation series. Therefore, compared to changes in SWP frequency, changes in SWP intensity contributed more to variations in $O_3$ concentration. To accurately assess the impact of both SWP frequency and intensity on annual $O_3$ variation, we quantitatively calculated their contributions. The contribution index was defined as the ratio of the interannual variation amplitude of the reconstructed series to that of the original series, i.e., ($O_{3max}$ of the reconstructed series − $O_{3min}$ of the reconstructed series) / ($O_{3max}$ of the original series − $O_{3min}$ of the original series). When only changes in SWP frequency were considered, their contribution to the interannual variation was 10.05%. When changes in SWP intensity were additionally included, the contribution increased to 69.66%. This indicates that, compared with changes in SWP frequency, changes in SWP intensity played a more important role in driving interannual variations in $O_3$ concentrations."

9. Lines 393–394: Contradiction between SWP1's role here and in Table 1, where it corresponds to low $O_3$ levels. This undermines the credibility of classification.

**Response:** Thank you for your insightful comment. Regarding the apparent contradiction between the role of SWP1 in Lines 393–394 and its association with low $O_3$ levels in Table 1, this is due to insufficient clarity in our description. The earlier part of the manuscript presents a preliminary analysis of weather patterns and $O_3$ pollution based on annual data. Subsequently, we performed a reclassification and focused

analysis of common weather patterns during the warm season (April to September) in the Yangtze River Delta, evaluating their contributions and future changes to the interannual variation of $O_3$ concentrations. Therefore, the definition and role of SWP1 differ between the annual and warm-season analyses, and these are not contradictory. We will clarify this distinction in the revised manuscript to avoid confusion.

In the revised manuscript, we unified the study period and selected June to September for SWP classification in the historical period (2018-2022) and future scenarios (SSP2-4.5 and SSP5-8.5). Each SWP described in the revised manuscript corresponds to each other to avoid misunderstanding.

10. Figure 1 caption: Missing units and variable definitions.

**Response:** Thank you for the valuable suggestion. We have added units and variable definitions in the caption of Figure 1 in the revised manuscript to improve its completeness and clarity.

Changes in the revised manuscript (lines 229–235) are shown as follows:

[Figure]

**Figure 1.** Inter-monthly variations in mean $O_3$ concentrations and associated meteorological conditions in the YRD from 2018 to 2022. **(a)** The monthly variation of $O_3$ concentrations is shown, with the red trend line representing the monthly mean values and the individual monthly concentrations labeled. **(b-c)** The variation trends of solar radiation, temperature, precipitation and relative humidity during the same period. (The shaded area in the violin plot represents the kernel density curve, with the three black lines indicating the 25th percentile, the mean, and the 75th percentile, respectively.)

11. Figures 3 & 4: The presentation of SWPs is unclear. Color bars and annotations should be enhanced for readability. Distinctions among SWPs are not evident visually.

**Response:** Thank you for your insightful suggestions. We have revised the figures in the revised manuscript. Specifically, we optimized the color scales and labels to make the different SWPs visually distinct and enhanced the overall readability of the figures. Changes in the revised manuscript (lines 311–315 and 329–334) are shown as follows:

[Figure]

**Figure 3.** The average weather conditions in SWP1, SWP2 and SWP3, including an 850 hPa geopotential height field superimposed on wind field **(a-c)**, sea level pressure field superimposed on 1000 hPa wind field **(d-f)**, a 500 hPa geopotential height field superimposed on wind field **(g-i)**. In **(a)-(i)**, shading represents geopotential height and color vectors represent wind with temperature. The black frame in **(a)-(i)** includes the YRD.

[Figure]

**Figure 4.** The average weather conditions in SWP4, including an 850 hPa geopotential height field superimposed on wind field **(a)**, sea level pressure field superimposed on 1000 hPa wind field **(b)**, a 500 hPa geopotential height field superimposed on wind field **(c)**, height-latitude cross-sections of vertical velocity (unit: $10^{-2}$ Pa·s$^{-1}$) between 25°N and 40°N **(d)**. In **(a)-(c)**, shading represents geopotential height and color vectors represent wind with temperature. The black frame in **(a)-(c)** and the vertical line area in **(d)** includes the YRD.

12. Materials and Methods: Define the "daily 8h average O$_3$" precisely, including time windows.

**Response:** Thank you for the valuable comment. We have clarified the precise definition of the "daily 8h average O$_3$" in the Materials and Methods section, including the exact time windows used for calculation, to ensure accuracy and clarity.

Changes in the revised manuscript (lines 129–140) are shown as follows:

2.1 O$_3$ observation data

"The hourly pollutant monitoring data of 26 cities in the YRD used are derived

from the National Environmental Monitoring Center of China. The platform provides pollutant concentration data updated every hour. To better describe the level of $O_3$ pollution at the urban scale, the arithmetic mean of the pollutant concentrations at each monitoring station was used as the pollutant concentration of the city. $O_3$-8h represents the daily maximum 8-hour average $O_3$ concentration, which can more accurately characterize the long-term exposure to regional $O_3$ pollution. The daily maximum 8-hour average $O_3$ concentration is calculated as the highest average $O_3$ mixing ratio over any consecutive 8-hour period within a calendar day (00:00–23:59 local time). Specifically, 8-hour moving averages are computed for all possible consecutive 8-hour windows (e.g., 00:00–07:59, 01:00–08:59, ..., 16:00–23:59), and the maximum value among them is recorded as the daily $O_3$ concentration. Therefore, the $O_3$ data used in the analysis of this article are all based on the $O_3$-8h value, and the daily maximum 8-hour average concentration (unit: $\mu g/m^3$) is taken as the daily $O_3$ concentration."

Finally, we would like to thank the reviewers for their comments and suggestions, which improved the rigor of our work and increased the article's novelty. Once again, thank you for your time. We hope that these revisions, made based on your comments, effectively address all of your concerns regarding our manuscript.

---

## Author Response (AR1)

**Response to Reviewers**

Journal: Atmospheric Chemistry and Physics

Manuscript No.: egusphere-2025-2466

Title of Paper: The impact of tropical cyclones on regional ozone pollution and its

future trend in the Yangtze River Delta of China

Authors: Mengzhu Xi, Min Xie\*, Da Gao, Danyang Ma, Yi Luo, Lingyun

Feng, Shitong Chen, Shuxian Zhang

Dear Editor,

We would like to submit our revised manuscript entitled "The impact of tropical cyclones on regional ozone pollution and its future trend in the Yangtze River Delta of China" to *Atmospheric Chemistry and Physics*.

On behalf of my co-authors, we thank you for handling the peer review of our manuscript. We appreciate the time and effort put in by you and the referees in reviewing our manuscript. These constructive comments are all valuable for revising and improving our manuscript. We have carefully studied these comments and made correction as requested. In the point-by-point responses, the reviewer's comments are in black, the author's responses are in blue, and the changes made to the text are highlighted in red. The line numbers in the author's responses are obtained from the clean version of revised manuscript, in which all the revisions have been accepted.

Best regards,

Authors

**Anonymous referee #1:**

I have carefully reviewed the manuscript entitled "The impact of tropical cyclones on regional ozone pollution and its future trend in the Yangtze River Delta of China". While the topic is of potential interest and relevance, particularly in the context of climate change and regional air quality, I cannot recommend publication in its current form. The manuscript suffers from multiple major flaws in methodology, presentation, data interpretation, and referencing, which severely undermine its scientific credibility. I therefore recommend rejection, although a substantially revised version may be reconsidered as a new submission. My major and specific concerns are listed below.

Response: We thank the referee #1 for the constructive comments and suggestions, which are very helpful for improving the clarity and reliability of the manuscript. We modified the methods, reanalyzed the data, rewrite the sentences for better readability, and enhanced the clarity of figures, to eliminate the mentioned major flaws in methodology, presentation, data interpretation, and referencing. Our point-by-point responses to the specific comments are listed as below.

**Major concerns:**

1. The authors identify five synoptic weather patterns (SWPs) using a rotated principal component analysis (PTT method), but they do not provide adequate information on the physical characteristics of these SWPs or justify the use of different meteorological parameters and domains in defining their intensities (Lines 167–171, 259–264).

**Response:** We thank the reviewer for the valuable comment. In the revised manuscript, we have made the following clarifications and improvements.

1. On the physical characteristics of the SWPs (Lines 259–264):

In the original manuscript, to focus on the main topic, only the SWP related to typhoon were analyzed in detail. The physical characteristics of the SWPs were briefly described, but the presentation was not sufficiently clear. Considering the comment of anonymous referee #1, in the revised manuscript, we have provided a clearer description of the typical characteristics of each classified SWP in Section 3.2, including the associated meteorological factors (such as air temperature, air humidity,

and wind speed) and the O3 concentration characteristics under different SWPs. In addition, the atmospheric circulation structure at 1000 hPa, 850 hPa and 500 hPa are presented and analyzed in Figures 3 and 4, which further reveal the dynamical and thermodynamical characteristics of each SWP. These analyses together improve and complement the characterization of the physical features of the identified SWPs.

Changes in the revised manuscript (lines 253–276) are shown as follows:

**3.2 Main synoptic weather patterns in the YRD**

"Based on the analysis of O3 variation characteristics in the YRD, O3 concentrations were generally high from June to September, coinciding with the peak season of TC activity. We selected June to September during 2018–2022 as the research period and used the PTT weather classification method to classify the weather situation. SWPs in the YRD during this period were primarily divided into four categories. As shown in Table 1, SWP1 is the main SWP, occurring on 383 days and accounting for 62.79 % of the period. SWP2 and SWP4 followed, occurring on 81 and 80 days, respectively, with frequencies of 13.28 % and 13.11 %. SWP3 was less frequent, occurring on 10.82 % of days. Specifically, SWP1 was mainly influenced by the southwesterly flow introduced by the WPSH and the northeast China low, SWP2 by northwesterly flow introduced by a continental high and the northeast China low, SWP3 by southeasterly flow introduced by the WPSH, and SWP4 by northeasterly flow introduced by the WPSH and TCs.

Statistics of average O3 concentrations and meteorological factors under each SWP indicate that SWP4 had the lowest average O3 concentration (115.07 μg/m³), whereas SWP2 had the highest value (132.36 μg/m³). O3 concentrations during SWP1 and SWP3 were similar, at 123.81 μg/m³ and 123.28 μg/m³, respectively. However, significant differences in O3 concentrations were observed among SWPs during the same period, highlighting the strong influence of circulation patterns on O3 pollution levels. SWP1 exhibited higher temperatures (27.12 °C), favoring increased O3 concentrations. SWP2 was influenced by dry northwesterly airflow, exhibiting lower humidity (76.74 %) and slower wind speed (2.03 m/s), which hindered air pollutant dispersion and resulted in higher O3 concentrations. SWP3 was characterized by lower

humidity (78.53 %) and slower wind speed (2.11 m/s), which were key factors contributing to increased O3 concentration. Under SWP4, though TCs far from the coastline could lead to regional O3 pollution (in section 3.3.2), the landfalling TCs could bring strong winds and rainstorms, increasing humidity (80.03 %) and wind speed (2.64 m/s), which resulted in lower average O3 concentrations."

**2. On the justification of the selected intensity factors (Lines 167–171):**

We sincerely apologize that the original manuscript did not adequately explain the rationale for the choice of different meteorological parameters and spatial domains in defining SWP intensities. In previous studies, a single intensity factor was often used for multiple weather patterns; however, this sometimes led to weak correlations with the ozone interannual variation series of a specific pattern, which in turn significantly affected the subsequent reconstruction. In our revised analysis, we took into account the unique features of each weather pattern when defining its intensity factor, and we further verified the rationality of the selection by calculating the correlation coefficients between candidate intensity factors and the ozone interannual variation series under each pattern. In the revised manuscript, detailed explanations have been added in Section 3.4.1, thereby providing a stronger basis for the construction of SWP intensity indices.

Changes in the revised manuscript (lines 420–437) are shown as follows:

3.4.1 The role of changes in the intensity and frequency of SWPs in the reconstruction of the annual variation series of  $O_3$

"In reconstructing the time series of annual O3 concentrations, we found that the definition of the SWP intensity factor played a crucial role in the reconstruction. Previous studies reconstructed the annual O3 concentration series by defining the SWP intensity factor as either the regional mean sea level pressure or the regional minimum pressure (Hegarty et al., 2007; Liu et al., 2019). However, this definition showed poor correlation between the SWP intensity factor and the annual O3 concentration series for certain SWPs. Therefore, we defined the SWP intensity factor for each SWP based on its specific meteorological characteristics, selecting the maximum, minimum, and mean geopotential heights across nine zones, and evaluated its validity by calculating the

correlation coefficient with the annual O3 variation series (Table 3). For SWP1 and SWP2, the maximum geopotential heights in zone 7 and zone 2 were highly correlated with the annual O3 variation series. For SWP3 and SWP4, the minimum geopotential heights in zone 9 and zone 4 were highly correlated with the annual O3 variation series. The maximum geopotential height reflects regional wind speeds, which determine the amount of water vapor transported into the region. Compared with SWP1, SWP2 has a larger weather system scale, so the maximum geopotential height in zone 2 shows a stronger correlation with the O3 series than that in zone 7. For SWP4, the YRD was affected by TCs, and O3 concentrations were closely related to TC intensity. The minimum geopotential height in zone 4 reflects the TC intensity. When the SWP intensity factor was defined based on the unique meteorological characteristics of each SWP, the reconstructed series more accurately reflected the impact of changes in SWP intensity on O3 concentrations."

**Table 3.** Correlation coefficients between the O3 concentration time series and different SWP intensity factors under each SWP.

|      | Zone 1
(115°~135°E, 20°~40°N) |       |       | Zone 2
(90°~140°E, 20°~50°N) |       |       | Zone 3
(110°~130°E, 10°~40°N) |       |       |
|------|----------------------------------|-------|-------|---------------------------------|-------|-------|----------------------------------|-------|-------|
| Type |                                  |       |       |                                 |       |       |                                  |       |       |
|      | Mean                             | Max   | Min   | Mean                            | Max   | Min   | Mean                             | Max   | Min   |
| SWP1 | -0.62                            | -0.72 | -0.41 | -0.69                           | 0.26  | 0.45  | -0.59                            | -0.76 | 0.17  |
| SWP2 | -0.65                            | -0.87 | -0.45 | -0.47                           | -0.88 | -0.09 | -0.60                            | -0.84 | -0.53 |
| SWP3 | -0.41                            | -0.67 | 0.72  | 0.10                            | -0.17 | 0.43  | -0.11                            | -0.60 | 0.75  |
| SWP4 | 0.15                             | -0.32 | 0.36  | 0.26                            | 0.01  | 0.32  | 0.33                             | -0.38 | 0.49  |
| Type | Zone 4                           |       |       | Zone 5                          |       |       | Zone 6                           |       |       |
|      | (110°~130°E, 25°~40°N)           |       |       | (100°~120°E, 15°~35°N)          |       |       | (110°~120°E, 15°~35°N)           |       |       |
|      | Mean                             | Max   | Min   | Mean                            | Max   | Min   | Mean                             | Max   | Min   |
| SWP1 | -0.67                            | -0.80 | -0.02 | -0.33                           | 0.37  | 0.05  | -0.60                            | -0.82 | 0.10  |
| SWP2 | -0.47                            | -0.73 | -0.44 | -0.36                           | -0.59 | -0.19 | -0.48                            | -0.65 | -0.29 |
| SWP3 | -0.19                            | -0.60 | 0.46  | 0.34                            | 0.31  | 0.67  | 0.29                             | -0.29 | 0.85  |
| SWP4 | 0.31                             | -0.32 | 0.64  | 0.48                            | -0.02 | 0.55  | 0.47                             | 0.26  | 0.55  |
|      | Zone 7                           |       |       | Zone 8                          |       |       | Zone 9                           |       |       |
| Type | (110°~130°E, 20°~35°N)           |       |       | (115°~135°E, 30°~50°N)          |       |       | (110°~130°E, 20°~40°N)           |       |       |
|      | Mean                             | Max   | Min   | Mean                            | Max   | Min   | Mean                             | Max   | Min   |
| SWP1 | -0.53                            | -0.83 | 0.06  | -0.40                           | -0.73 | 0.42  | -0.61                            | -0.76 | 0.09  |
| SWP2 | -0.54                            | -0.79 | -0.25 | -0.31                           | -0.72 | 0.03  | -0.56                            | -0.77 | -0.50 |
| SWP3 | 0.02                             | -0.69 | 0.86  | -0.25                           | -0.64 | 0.33  | -0.08                            | -0.60 | 0.87  |

Furthermore, Table 1 suggests that some SWPs primarily occur in winter, spring, and autumn, while SWP2 and SWP5 dominate in summer. This implies a strong seasonal signal embedded in the classification. However, the authors proceed to analyses O3 pollution impacts without removing seasonal cycles, which is inappropriate given the strong seasonal variability in both O3 precursors and photochemistry. Weather pattern classification should be conducted on seasonally detrended data to avoid conflating synoptic and seasonal influences.

Response: Thank you for your meticulous review and valuable comments on our manuscript. We understand the concern regarding the potential influence of seasonal signals on the analysis of SWPs and O3. In the revised manuscript, we have taken this into consideration and provided a clear explanation. For a year-round analysis, to perform seasonally detrended analysis is indeed reasonable. However, the focus of this study is on the impact of tropical cyclones on O3 pollution. Our analysis shows that O3 pollution in the Yangtze River Delta predominantly occurs from June to September, which coincides with the peak period of tropical cyclone activity. Therefore, we have adjusted the study period to **June–September** and conducted weather pattern classification only for this period, excluding year-round data. This adjustment effectively minimizes the influence of seasonal variability on the analysis results. It also ensures that the classification is focused on the relevant conditions for assessing the impact of tropical cyclones on O3, reducing the confounding effects of seasonal changes on O3 precursors and photochemistry, and providing a more robust basis for evaluating the role of weather patterns in O3 pollution.

Changes in the revised manuscript (lines 253–263) are shown as follows:

**3.2 Main synoptic weather patterns in the YRD**

"Based on the analysis of O3 variation characteristics in the YRD, O3 concentrations were generally high from June to September, coinciding with the peak season of TC activity. We selected June to September during 2018–2022 as the research period and used the PTT weather classification method to classify the weather situation.

SWPs in the YRD during this period were primarily divided into four categories. As shown in Table 1, SWP1 is the main SWP, occurring on 383 days and accounting for 62.79 % of the period. SWP2 and SWP4 followed, occurring on 81 and 80 days, respectively, with frequencies of 13.28 % and 13.11 %. SWP3 was less frequent, occurring on 10.82 % of days. Specifically, SWP1 was mainly influenced by the southwesterly flow introduced by the WPSH and the northeast China low, SWP2 by northwesterly flow introduced by a continental high and the northeast China low, SWP3 by southeasterly flow introduced by the WPSH, and SWP4 by northeasterly flow introduced by the WPSH and TCs."

2. Many key numerical conclusions lack both graphical support and proper referencing (e.g., Lines 394–404, 436–438). For instance, the stated contributions of SWP5 to O3 concentration changes—15.14% in 2030 and 20.66% in 2060—are not clearly contextualized. These values appear to indicate that SWP5 is the most influential within those respective years compared to other years, rather than compared to other weather patterns. However, the manuscript does not clearly define the comparison baseline, which may mislead readers into interpreting these percentages as reflecting the dominance of SWP5 over other synoptic types. Clarification is essential to avoid misinterpretation.

**Response:** We thank the reviewer for this careful comment. We sincerely apologize that some numerical conclusions in the original manuscript lacked graphical support and were not clearly presented, which could potentially lead to misinterpretation. In the revised manuscript, we have clarified and improved these points as follows.

- 1. To address the issue of key numerical conclusions lacking graphical support, we have updated the relevant figures in the revised manuscript to provide visual support for these results (e.g., Figure 7). These updates allow readers to more clearly understand the numerical values and enhance the visualization and interpretability of the conclusions. Changes in the revised manuscript (lines 456–465) are shown as follows:
- 3.4.2 Changes in TCs and the effects on O3 concentration over the YRD in the future

"Under the SSP2-4.5 scenario, the frequencies of SWP1 to SWP4 were 57.79%, 19.67%, 6.76%, and 15.78%, respectively (Fig. 10). High-average O3 pattern occurred on 93, 99, 96, and 90 days in 2030, 2035, 2060, and 2100, respectively, showing a general increasing trend, with the peak occurring in 2035. These results suggest that, considering only changes in SWP frequency, O3 concentrations would reach a maximum in 2035 under the SSP2-4.5 scenario. Under the SSP5-8.5 scenario, the frequencies of SWP1 to SWP4 were 50.82%, 23.98%, 7.17%, and 18.03%, respectively (Fig. 10). High-average O3 pattern will occur on 99, 90, 82, and 94 days in 2030, 2035, 2060, and 2100, respectively, under the SSP5-8.5 scenario, with the number of days increasing in 2030 and 2100. This also suggests that, considering only changes in SWP frequency, O3 concentrations would increase in 2030 and 2100 under the SSP5-8.5 scenario."

**Figure 7.** Distribution of the number of days of occurrence of each SWP in the YRD from June to September under the historical period (2018-2022), SSP2-4.5, and SSP5-8.5 future scenarios.

2. The reported percentages (e.g., 15.14% in 2030 and 20.66% in 2060) represent the relative contributions of each SWP to O3 changes within the same year, rather than direct comparisons across different weather patterns. We have revised the wording in the manuscript to make it clear that historical tropical cyclone contributions serve as

the comparison baseline, thereby preventing potential misinterpretation. To further enhance the visualization and interpretability, we have added the relevant figure (Figure 10b) and included necessary references in the text, allowing readers to intuitively understand the contributions of each weather pattern to O3 changes. These revisions ensure that the conclusions are clearly presented and readily understandable.

Changes in the revised manuscript (lines 514–528) are shown as follows:

3.4.3 Reconstruction of annual O3 variability in the YRD under future scenarios

"Figure 10a shows the contribution of each SWP to the reconstructed series of future O3 concentrations in the YRD from June to September under the SSP2-4.5 and SSP5-8.5 scenarios. We focus on the contribution of the TC weather pattern (SWP4) to the reconstructed O3 concentration series under different scenarios (Figure 10b). During the historical period (2018–2022), the TC weather pattern contributed an average of 13.11% to O3 concentrations in the YRD. Under the SSP2-4.5 scenario, the contributions of the TC weather pattern in 2030, 2035, 2060, and 2100 were 15.57%, 13.93%, 17.21%, and 16.39%, respectively. The frequency of the TC weather pattern in the YRD in 2035 was lower, resulting in a lower contribution of the TC weather pattern to O3 concentrations in that year. Under the SSP5-8.5 scenario, the TC weather pattern contributes 14.75%, 15.57%, 24.59%, and 19.67% to O3 changes in 2030, 2035, 2060, and 2100, respectively. Under both the SSP2-4.5 and SSP5-8.5 scenarios, the contribution of the TC weather pattern to O3 pollution increases to varying degrees compared with the historical period. This suggests that under future climate change, the impact of TC weather pattern on O3 pollution in the YRD may intensify. Further research is needed on the relationship between key SWPs, such as TC weather pattern, and O3 formation mechanisms to more accurately predict and mitigate regional O3 pollution under future climate conditions."

**Figure 10.** Contribution of SWP to the projected annual variation series of O3 concentration under the SSP2-4.5 and SSP5-8.5 future scenarios. (a) Contribution of four SWPs, (b) Contribution of TC weather pattern (SWP4).

Several critical results, such as the contribution index and the role of TC-generated meteorological conditions (e.g., "low humidity and strong solar radiation"), lack quantitative evidence (Lines 309–311). This weakens the entire discussion on the role of TCs in driving O3 changes.

**Response:** We thank the reviewer for this insightful comment. Regarding the problem that "the contribution index and the role of TC-generated meteorological conditions lack quantitative evidence", we have provided clarifications and additional details in the revised manuscript.

1. We recognize that the original manuscript on lines 309–311 may have caused misunderstanding, implying that "tropical cyclones directly cause low humidity and strong solar radiation". In fact, our intention was to describe the summer meteorological conditions themselves—characterized by high temperature, low humidity, and strong radiation—which provide favorable conditions for O3 formation, rather than being directly induced by tropical cyclones. To avoid such misinterpretation, we have revised the relevant text in the revised manuscript and analyzed the physical characteristics of each weather pattern during the selected study period, clearly distinguishing the effects of tropical cyclone—induced changes on O3.

Changes in the revised manuscript (lines 311–321) are shown as follows:

**3.3.1 The impact of TC weather pattern (SWP4) on O3 pollution in the YRD**

"Figure 4 illustrates the three-dimensional atmospheric circulation structure under the TC weather pattern (SWP4). For the TC weather pattern, similar circulation conditions were observed at 850 hPa (Fig. 4a) and at sea level (Fig. 4b). The YRD was located northwest of the TCs and was controlled by northeasterly flow guided by the TCs. The direction and intensity of the northeast wind had a significant impact on meteorological conditions and pollutant transport in the YRD. At 500 hPa, the region was dominated by westerly or northwesterly flow (Fig. 4c). Meanwhile, the peripheral downward airflow associated with lower-level TCs (Fig. 4d) led to a more stagnant atmosphere over the YRD. As the TC approaches the YRD, strong northeasterly flow increased clean sea airflow transportation to the region, lowered temperatures and increased humidity, creating unfavorable meteorological conditions for photochemical reactions. Furthermore, higher wind speeds facilitated air pollutant elimination, leading to a decrease in O3 concentrations in the region (Table 1)."

2. The description of the contribution index in the original manuscript was not sufficiently clear, leading to ambiguity in its mathematical definition and physical interpretation. To accurately quantify the impacts of changes in SWP frequency and intensity on interannual O3 variations, we calculated the contribution index as the ratio of the interannual amplitude of the reconstructed sequence (maximum minus minimum O3 concentration within the reconstructed sequence) to that of the original sequence (maximum minus minimum O3 concentration in the original series). This index represents the proportion of interannual O3 variation explained by a given factor (e.g., changes in SWP frequency or combined frequency and intensity). The results indicate that when considering only SWP frequency changes, the reconstructed contribution is 10.05%, whereas including SWP intensity changes increases the reconstructed contribution to 69.66%, demonstrating that intensity variations have a much stronger influence on interannual O3 differences than frequency variations. We have revised the manuscript accordingly to make the definition, calculation, and physical meaning of the contribution index clear to readers.

Changes in the revised manuscript (lines 401–414) are shown as follows:

3.4.1 The role of changes in the intensity and frequency of SWPs in the reconstruction of the annual variation series of  $O_3$

"When only changes in SWP frequency were considered, the reconstructed time series was relatively flat and did not adequately capture the variation trend of O3 concentrations. Therefore, changes in SWP frequency had minimal impact on the annual variation of O3. When changes in SWP intensity were considered, the reconstructed series more closely resembled the original annual variation series. Therefore, compared to changes in SWP frequency, changes in SWP intensity contributed more to variations in O3 concentration. To accurately assess the impact of both SWP frequency and intensity on annual O3 variation, we quantitatively calculated their contributions. The contribution index was defined as the ratio of the interannual variation amplitude of the reconstructed series to that of the original series, i.e.,  $(O_{3\text{max}})$ of the reconstructed series –  $O_{3min}$  of the reconstructed series) /  $(O_{3max}$  of the original series - O3min of the original series). When only changes in SWP frequency were considered, their contribution to the interannual variation was 10.05%. When changes in SWP intensity were additionally included, the contribution increased to 69.66%. This indicates that, compared with changes in SWP frequency, changes in SWP intensity played a more important role in driving interannual variations in O3 concentrations."

3. Throughout the manuscript—particularly in the introduction—numerous claims are either incorrectly cited, misrepresented, or not supported by the referenced sources. This significantly undermines the scientific credibility and rigour of the work.

Response: Thank you very much for your careful review of our research and your valuable feedback. We are deeply sorry for the issues you pointed out, including inappropriate citations, misinterpretations, and lack of supporting references. We attach great importance to this and have carefully reviewed and analyzed the manuscript. During this process, we found that some of the issues may be due to unclear presentation or misunderstandings. To address these misunderstandings, we have reorganized and revised the relevant paragraphs, striving for greater clarity and logic. We believe that these revisions better convey our views and research ideas while

enhancing the scientific rigor of our research.

Here are some examples, lines 38–40: The statement that O3 accounted for over 50% of polluted days is not substantiated by the referenced China Ecological Environment Bulletin. The authors should provide the correct citation, improve the clarity of the English, and revise the conclusion to reflect the actual content of the report.

**Response:** Thank you for pointing this out. After checking, we confirm that the phrase "in many Chinese cities" in the original manuscript was inaccurate. The data we cited are from the 2020 China Ecological Environment Bulletin for the Yangtze River Delta region, where O3 was the primary pollutant on 50.7% of the total number of polluted days. We have modified the sentence in the revised manuscript.

Changes in the revised manuscript (lines 40–42) are shown as follows:

"...According to the 2020 China Ecological Environment Bulletin, in the YRD, the number of days with air quality exceeding the standard with O3 as the primary pollutant accounted for over 50% of the total number of days exceeding the standard." Lines 77–80: Assertions regarding the low-frequency modulation of tropical cyclones (TCs) by climate change are made without any credible or peer-reviewed references.

Response: Thank you for your valuable comment. We sincerely apologize for not providing relevant references earlier. Thank you for your valuable comment. We have now included authoritative peer-reviewed literature to strengthen the scientific basis of our discussion regarding the low-frequency modulation of tropical cyclones by climate

Changes in the revised manuscript (lines 79–83) are shown as follows:

"...In the context of global climate change, with the increases in greenhouse gas emissions, global warming, and interannual climate changes, the state of the atmosphere and ocean has changed, and sea-level rise and extreme climate events have occurred frequently, which have a low-frequency modulating effect on TCs (Chu et al., 2020; Moon et al., 2023; Moon et al., 2025; Wang et al., 2023a) ..."

**References**

change.

Chu, J. E., Lee, S. S., Timmermann, A., Wengel, C., Stuecker, M. F., and Yamaguchi,

R.: Reduced tropical cyclone densities and ocean effects due to anthropogenic greenhouse warming, Science Advances, 6, <a href="http://doi.org/10.1126/sciadv.abd5109">http://doi.org/10.1126/sciadv.abd5109</a>, 2020.

Moon, M., Ha, K. J., Kim, D., Ho, C. H., Park, D. S. R., Chu, J. E., Lee, S. S., and Chan, J. C. L.: Rainfall strength and area from landfalling tropical cyclones over the North Indian and western North Pacific oceans under increased CO2 conditions, Weather and Climate Extremes, 41, <a href="http://doi.org/10.1016/j.wace.2023.100581">http://doi.org/10.1016/j.wace.2023.100581</a>, 2023.

Moon, M., Min, S. K., Chu, J. E., An, S. I., Son, S. W., Ramsay, H., and Wang, Z.: Tropical cyclone response to ambitious decarbonization scenarios, Npj Climate and Atmospheric Science, 8, <a href="http://doi.org/10.1038/s41612-025-01122-9">http://doi.org/10.1038/s41612-025-01122-9</a>, 2025.

Wang, S., Murakami, H., and Cooke, W. F.: Anthropogenic forcing changes coastal tropical cyclone frequency, Npj Climate and Atmospheric Science, 6, <a href="http://doi.org/10.1038/s41612-023-00516-x">http://doi.org/10.1038/s41612-023-00516-x</a>, 2023a.

Lines 86–88: The claim that TC frequency in the Northwest Pacific has decreased, while intensity and duration have increased, is not convincingly supported by the cited literature. In fact, some references appear to be news media or secondary summaries rather than original peer-reviewed scientific studies.

Response: Thank you very much for your careful review of our manuscript and your valuable comments. Regarding the references on changes in the frequency, intensity, and duration of tropical cyclones in the Northwest Pacific, we have carefully reexamined the relevant literature to avoid any possible misunderstanding. We would like to clarify that all the references cited in the manuscript are peer-reviewed scientific articles rather than news media or secondary summaries. In the original manuscript, Yu et al. (2025) was cited to support the view that "tropical cyclones in the western Pacific are becoming more intense". However, this article does not directly examine changes in tropical cyclone intensity. Instead, its introduction provides a broader context for the increasing intensity of tropical cyclones in the western Pacific. Therefore, we have replaced it with a more appropriate reference. Recent peer-reviewed studies (e.g.,

(Balaguru et al., 2024; Bhatia et al., 2022; Chand et al., 2022; Jung, 2025; Wang et al., 2022a; Kossin, 2018; Zhang et al., 2020) indicate that under global warming, the frequency of tropical cyclones in the western Pacific is decreasing, but their average intensity is increasing, and their duration is also increasing.

Changes in the revised manuscript (lines 89–92) are shown as follows:

"...Although the frequency of TCs in the northwest Pacific Ocean has decreased in recent years, their average intensity has shown an upward trend (Balaguru et al., 2024; Bhatia et al., 2022; Chand et al., 2022; Jung, 2025; Wang et al., 2022a) and their duration has also become longer (Kossin, 2018; Zhang et al., 2020)..."

**References**

- Balaguru, K., Chang, C. C., Leung, L. R., Foltz, G. R., Hagos, S. M., Wehner, M. F., Kossin, J. P., Ting, M. F., and Xu, W. W.: A Global Increase in Nearshore Tropical Cyclone Intensification, Earths Future, 12, http://doi.org/10.1029/2023ef004230, 2024.
- Bhatia, K., Baker, A., Yang, W. C., Vecchi, G., Knutson, T., Murakami, H., Kossin, J., Hodges, K., Dixon, K., Bronselaer, B., and Whitlock, C.: A potential explanation for the global increase in tropical cyclone rapid intensification, Nature Communications, 13, http://doi.org/10.1038/s41467-022-34321-6, 2022.
- Chand, S. S., Walsh, K. J. E., Camargo, S. J., Kossin, J. P., Tory, K. J., Wehner, M. F., Chan, J. C. L., Klotzbach, P. J., Dowdy, A. J., Bell, S. S., Ramsay, H. A., and Murakami, H.: Declining tropical cyclone frequency under global warming, Nature Climate Change, 12, 655-+, http://doi.org/10.1038/s41558-022-01388-4, 2022.
- Jung, H.: Humans fuel stronger cyclones, Nature Climate Change, 15, 351-351, http://doi.org/10.1038/s41558-025-02321-1, 2025.
- Kossin, J. P.: A global slowdown of tropical-cyclone translation speed, Nature, 558, 104-+, http://doi.org/10.1038/s41586-018-0158-3, 2018.
- Wang, G. H., Wu, L. W., Mei, W., and Xie, S. P.: Ocean currents show global intensification of weak tropical cyclones, Nature, 611, 496-+, http://doi.org/10.1038/s41586-022-05326-4, 2022a.

Zhang, G., Murakami, H., Knutson, T. R., Mizuta, R., and Yoshida, K.: Tropical cyclone motion in a changing climate, Science Advances, 6, http://doi.org/10.1126/sciadv.aaz7610, 2020.

**Specific Comments**

1. Line 50: Please spell out "TC" at first mention.

**Response:** Thank you for pointing this out. We have spelled out "TC" at its first occurrence as "tropical cyclone (TC)" and retained the abbreviation consistently thereafter.

Changes in the revised manuscript (lines 52–53) are shown as follows:

"Tropical cyclone (TC) activities have a profound impact on the ecological environment of China's coastal areas. ..."

2. Lines 190–192: Claims on O3 variation lack data or plots. Add evidence.

Response: Thank you for your careful review of our manuscript and your scientific suggestions. According to your suggestions, we added this part to the revised manuscript, enhancing the rationality and completeness of our analysis. We have also modified and optimized Figure 1, adding trend lines to more clearly illustrate the trend of O3 concentration. Key data points, such as the peaks and valleys of O3 concentration, are detailed in the figure, with the corresponding concentration values highlighted at peak locations for easier understanding. Furthermore, based on the original figure, we have added temporal trends in meteorological factors such as solar radiation, temperature, precipitation, and relative humidity to further support our assertion that meteorological conditions drive O3 changes.

Changes in the revised manuscript (lines 222–228) are shown as follows:

**Figure 1.** Inter-monthly variations in mean O3 concentrations and associated meteorological conditions in the YRD from 2018 to 2022. **(a)** The monthly variation of O3 concentrations is shown, with the red trend line representing the monthly mean values and the individual monthly concentrations labeled. **(b-c)** The variation trends of solar radiation, temperature, precipitation and relative humidity during the same period. (The shaded area in the violin plot represents the kernel density curve, with the three black lines indicating the 25th percentile, the mean, and the 75th percentile, respectively.)

3. Lines 213–217: The description of two TC-O3 response types is unsupported. Provide composite analysis and classify examples accordingly.

Response: We thank the reviewer for this valuable comment. Following your suggestion, we have clarified the relevant section in the revised manuscript. In particular, Figure 2 provides a clear visualization of the O3 concentration changes during tropical cyclone events, where two typical patterns can be observed: one showing an increase–decrease sequence, and the other showing an increase–decrease–increase sequence. These trends indicate that tropical cyclones indeed influence O3 concentrations, providing a theoretical basis for our subsequent analyses. The main purpose of this section was to highlight the impact of tropical cyclones on O3, rather than to conduct an in-depth exploration of specific "TC-O3 response types". Therefore, we have simplified and clarified the wording in the revised manuscript to avoid potential ambiguity.

Changes in the revised manuscript (lines 229–247) are shown as follows:

**3.1 Characteristics of O3 pollution in the YRD**

"O3 pollution in the YRD is concentrated in warm months of the year. We analyzed the temporal distribution of O3 concentrations in representative cities in the YRD (Shanghai, Nanjing, Hangzhou, and Hefei) from April to September during 2018–2022 (Fig. 2). Since TCs mainly occur after June, O3 pollution is often associated with TC activity. During the same TC period, the O3 concentrations in Shanghai, Nanjing, Hangzhou, and Hefei were different, but the temporal trends were similar. TCs have a certain impact on the changes in O3 concentrations in the YRD. According to the evolution and trajectory of TC weather, TCs affecting China are primarily generated in the northwest Pacific Ocean (Zhan et al., 2020; Wang et al., 2024a). As they develop and move, these TCs will have a significant impact on O3 concentrations in China. At this time, the YRD is located on the periphery of the TCs. Under the control of the periphery of the TCs, the strong downward airflow will make the YRD in stagnant weather conditions and inhibit the diffusion of pollutants (Shu et al., 2016), accompanied by high temperature, clear and dry weather conditions, which are the main weather conditions causing O3 pollution. With the evolution of TC weather system, when the TCs center gradually approaches the YRD until it is within a certain range, the YRD is no longer under the influence of the periphery of the TCs and comes under the control of the TC wind and rain belt. Strong winds and precipitation can significantly cleanse air pollutants, and thereby reduce O3 concentrations. After TCs dissipate, O3 pollution levels may rise again due to restored meteorological conditions conducive to O3 formation (Zhan et al., 2020). These results suggest that TCs can significantly affect O3 pollution in the YRD, highlighting the importance of implementing O3 control measures prior to TC landfall."

4. Lines 217–218: No references for TC genesis location or influence radius. Clarify language: "generation" implies early-stage oceanic development, likely too far from YRD to affect local O3.

**Response:** We thank the reviewer for this helpful comment. We acknowledge that the original manuscript in Lines 217–218 may cause ambiguity. Our intention was to state

that the tropical cyclones affecting the YRD mainly originate in the northwest Pacific and subsequently influence the regional atmosphere through their development and trajectories, rather than implying that their early-stage oceanic genesis directly affects local O3. To avoid confusion, we have revised the wording in the manuscript and added appropriate references (e.g., Zhan et al., 2020; Wang et al., 2024) to support the description of the typical genesis locations and influence range of TCs.

Changes in the revised manuscript (lines 234–241) are shown as follows:

- 3.1 Characteristics of O3 pollution in the YRD
- "...According to the evolution and trajectory of TC weather, TCs affecting China are primarily from the northwest Pacific Ocean (Zhan et al., 2020; Wang et al., 2024a). As they develop and move, these TCs will have a significant impact on O3 concentrations in China (Xi et al., 2025). At this time, the YRD is located on the periphery of the TCs. Under the control of the periphery of the TCs, the strong downward airflow will make the YRD in stagnant weather conditions and inhibit the diffusion of pollutants (Shu et al., 2016), accompanied by high temperature, clear and dry weather conditions, which are the main weather conditions causing O3 pollution ..."

**References**

- Wang, J. H., Wang, P., Tian, C. F., Gao, M., Cheng, T. T., and Mei, W.: Consecutive Northward Super Typhoons Induced Extreme Ozone Pollution Events in Eastern China, Npj Climate and Atmospheric Science, 7, <a href="http://doi.org/10.1038/s41612-024-00786-z">http://doi.org/10.1038/s41612-024-00786-z</a>, 2024a.
- Xi, M., Luo, Y., Li, Y., Ma, D., Feng, L., Zhang, S., Chen, S., and Xie, M.: Comprehensive analysis of prevailing weather patterns and high-impact typhoon tracks to reveal where and how tropical cyclone affects regional ozone pollution in the Yangtze River Delta region, China, Atmospheric Environment, 361, 121498, http://doi.org/https://doi.org/10.1016/j.atmosenv.2025.121498, 2025.
- Zhan, C. C., Xie, M., Huang, C. W., Liu, J. N., Wang, T. J., Xu, M., Ma, C. Q., Yu, J.
  W., Jiao, Y. M., Li, M. M., Li, S., Zhuang, B. L., Zhao, M., and Nie, D. Y.: Ozone affected by a succession of four landfall typhoons in the Yangtze River Delta, China: major processes and health impacts, Atmospheric Chemistry and Physics,

5. Lines 227–229: Claims not supported by figures. Add case studies or remove.

**Response:** Thank you for the reviewer's comments. Although these claims are supported by figures and data in a later section of our manuscript, it is not directly supported by the data or figures presented in the context of Lines 227–229. To maintain logical rigor and avoid potential misunderstanding at this stage, we have removed the claims from this section.

6. Lines 309–311: The link between TC conditions and "low humidity, strong solar radiation" is unsubstantiated. TCs often increase regional moisture. Provide observational support.

Response: Thank you for the reviewer's insightful comments on our manuscript. We recognize that the original manuscript in Lines 309–311 may have caused misunderstanding, implying that "tropical cyclones directly cause low humidity and strong solar radiation". In fact, our intention was to describe the summer meteorological conditions themselves—characterized by high temperature, low humidity, and strong radiation—which provide favorable conditions for O3 formation, rather than being directly induced by tropical cyclones. To avoid such misinterpretation, we have revised the relevant text in the manuscript and analyzed the physical characteristics of each weather pattern during the selected study period, clearly distinguishing the effects of tropical cyclone—induced changes on O3.

Changes in the revised manuscript (lines 311–321) are shown as follows:

3.3.1 The impact of TC weather pattern (SWP4) on O3 pollution in the YRD

"Figure 4 illustrates the three-dimensional atmospheric circulation structure under the TC weather pattern (SWP4). For the TC weather pattern, similar circulation conditions were observed at 850 hPa (Fig. 4a) and at sea level (Fig. 4b). The YRD was located northwest of the TCs and was controlled by northeasterly flow guided by the TCs. The direction and intensity of the northeast wind had a significant impact on meteorological conditions and pollutant transport in the YRD. At 500 hPa, the region

was dominated by westerly or northwesterly flow (Fig. 4c). Meanwhile, the peripheral downward airflow associated with lower-level TCs (Fig. 4d) led to a more stagnant atmosphere over the YRD. As the TC approaches the YRD, strong northeasterly flow increased clean sea airflow transportation to the region, lowered temperatures and increased humidity, creating unfavorable meteorological conditions for photochemical reactions. Furthermore, higher wind speeds facilitated air pollutant elimination, leading to a decrease in O3 concentrations in the region (Table 1)."

7. Line 348: The 500 hPa ridge discussion (Fig. 5g–i) is unclear. Differences between panels are not visually apparent. Consider replotting with clearer contrasts.

**Response:** Thank you for pointing this out. In response to the concern that the discussion of the 500 hPa ridge line (Fig. 5g-i) was unclear and that the differences were difficult to distinguish, we have redefined the study periods and redrawn the corresponding figures in the revised manuscript. These modifications highlight the contrasts more effectively and make the results clearer and easier to interpret.

8. Lines 368–372: The "contribution index" needs a mathematical definition and justification. The interpretation lacks physical meaning without context.

Response: Thank you for your valuable comment. The description of the contribution index in the original manuscript was not sufficiently clear, leading to ambiguity in its mathematical definition and physical interpretation. In this study, to accurately quantify the impacts of changes in SWP frequency and intensity on interannual O3 variations, we calculated the contribution index as the ratio of the interannual amplitude of the reconstructed sequence (maximum minus minimum O3 concentration within the reconstructed sequence) to that of the original sequence (maximum minus minimum O3 concentration in the original series). This index represents the proportion of interannual O3 variation explained by a given factor (e.g., changes in SWP frequency or combined frequency and intensity). The results indicate that when considering only SWP frequency changes, the reconstructed contribution is 10.05%, whereas including SWP intensity changes increases the reconstructed contribution to 69.66%, demonstrating

that intensity variations have a much stronger influence on interannual O3 differences than frequency variations. We have revised the manuscript accordingly to make the definition, calculation, and physical meaning of the contribution index clear to readers. Changes in the revised manuscript (lines 401–414) are shown as follows:

3.4.1 The role of changes in the intensity and frequency of SWPs in the reconstruction of the annual variation series of O3

"When only changes in SWP frequency were considered, the reconstructed time series was relatively flat and did not adequately capture the variation trend of O3 concentrations. Therefore, changes in SWP frequency had minimal impact on the annual variation of O3. When changes in SWP intensity were considered, the reconstructed series more closely resembled the original annual variation series. Therefore, compared to changes in SWP frequency, changes in SWP intensity contributed more to variations in O3 concentration. To accurately assess the impact of both SWP frequency and intensity on annual O3 variation, we quantitatively calculated their contributions. The contribution index was defined as the ratio of the interannual variation amplitude of the reconstructed series to that of the original series, i.e., (O3max of the reconstructed series –  $O_{3min}$  of the reconstructed series) /  $(O_{3max}$  of the original series - O3min of the original series). When only changes in SWP frequency were considered, their contribution to the interannual variation was 10.05%. When changes in SWP intensity were additionally included, the contribution increased to 69.66%. This indicates that, compared with changes in SWP frequency, changes in SWP intensity played a more important role in driving interannual variations in O3 concentrations."

9. Lines 393–394: Contradiction between SWP1's role here and in Table 1, where it corresponds to low O3 levels. This undermines the credibility of classification.

**Response:** Thank you for your insightful comment. Regarding the apparent contradiction between the role of SWP1 in Lines 393–394 and its association with low O3 levels in Table 1, this is due to insufficient clarity in our description. The earlier part of the manuscript presents a preliminary analysis of weather patterns and O3 pollution based on annual data. Subsequently, we performed a reclassification and focused

analysis of common weather patterns during the warm season (April to September) in the Yangtze River Delta, evaluating their contributions and future changes to the interannual variation of O3 concentrations. Therefore, the definition and role of SWP1 differ between the annual and warm-season analyses, and these are not contradictory. We will clarify this distinction in the revised manuscript to avoid confusion.

In the revised manuscript, we unified the study period and selected June to September for SWP classification in the historical period (2018-2022) and future scenarios (SSP2-4.5 and SSP5-8.5). Each SWP described in the revised manuscript corresponds to each other to avoid misunderstanding.

**10. Figure 1 caption: Missing units and variable definitions.**

**Response:** Thank you for the valuable suggestion. We have added units and variable definitions in the caption of Figure 1 in the revised manuscript to improve its completeness and clarity.

Changes in the revised manuscript (lines 222–228) are shown as follows:

**Figure 1.** Inter-monthly variations in mean O3 concentrations and associated meteorological conditions in the YRD from 2018 to 2022. **(a)** The monthly variation of O3 concentrations is shown, with the red trend line representing the monthly mean values and the individual monthly concentrations labeled. **(b-c)** The variation trends of solar radiation, temperature, precipitation and relative humidity during the same period. (The shaded area in the violin plot represents the kernel density curve, with the three black lines indicating the 25th percentile, the mean, and the 75th percentile, respectively.)

11. Figures 3 & 4: The presentation of SWPs is unclear. Color bars and annotations should be enhanced for readability. Distinctions among SWPs are not evident visually. **Response:** Thank you for your insightful suggestions. We have revised the figures in the revised manuscript. Specifically, we optimized the color scales and labels to make the different SWPs visually distinct and enhanced the overall readability of the figures. Changes in the revised manuscript (lines 304–308 and 322–327) are shown as follows:

**Figure 3.** The average weather conditions in SWP1, SWP2 and SWP3, including an 850 hPa geopotential height field superimposed on wind field (**a-c**), sea level pressure field superimposed on 1000 hPa wind field (**d-f**), a 500 hPa geopotential height field superimposed on wind field (**g-i**). In (**a**)-(**i**), shading represents geopotential height and color vectors represent wind with temperature. The black frame in (**a**)-(**i**) includes the YRD.

**Figure 4.** The average weather conditions in SWP4, including an 850 hPa geopotential height field superimposed on wind field (a), sea level pressure field superimposed on 1000 hPa wind field (b), a 500 hPa geopotential height field superimposed on wind field (c), height-latitude cross-sections of vertical velocity (unit:  $10^{-2} \text{ Pa} \cdot \text{s}^{-1}$ ) between 25°N and 40°N (d). In (a)-(c), shading represents geopotential height and color vectors represent wind with temperature. The black frame in (a)-(c) and the vertical line area in (d) includes the YRD.

12. Materials and Methods: Define the "daily 8h average O3" precisely, including time windows.

**Response:** Thank you for the valuable comment. We have clarified the precise definition of the "daily 8h average O3" in the Materials and Methods section, including the exact time windows used for calculation, to ensure accuracy and clarity.

Changes in the revised manuscript (lines 122–133) are shown as follows:

2.1 O3 observation data

"The hourly pollutant monitoring data of 26 cities in the YRD used are derived

from the National Environmental Monitoring Center of China. The platform provides pollutant concentration data updated every hour. To better describe the level of O3 pollution at the urban scale, the arithmetic mean of the pollutant concentrations at each monitoring station was used as the pollutant concentration of the city. O3-8h represents the daily maximum 8-hour average O3 concentration, which can more accurately characterize the long-term exposure to regional O3 pollution. The daily maximum 8-hour average O3 concentration is calculated as the highest average O3 mixing ratio over any consecutive 8-hour period within a calendar day (00:00–23:59 local time). Specifically, 8-hour moving averages are computed for all possible consecutive 8-hour windows (e.g., 00:00–07:59, 01:00–08:59, ..., 16:00–23:59), and the maximum value among them is recorded as the daily O3 concentration. Therefore, the O3 data used in the analysis of this article are all based on the O3-8h value, and the daily maximum 8-hour average concentration (unit: μg/m3) is taken as the daily O3 concentration."

**Anonymous referee #2:**

General comments: The manuscript investigates the impact of tropical cyclones on ozone and its future changing trends in the Yangtze River Delta region. Their results shows that the regional O3 pollution are affected by tropical cyclones, and based on future scenario data, they further discuss the changes in tropical cyclones and their impact on the trend of O3. The results are interesting for understanding the future change of tropical cyclones and O3. Several points of the manuscript still need to be improved before accepted. Specifically, they show that the regional O3 pollution usually occurred before and after tropical cyclones in 2018–2022 as shown in their analysis, why do they use the data reconstruction about the TC weather patterns to evaluate the annual change of O3 concentrations in the future, instead of extreme O3 pollution? I believe that the evaluation of extreme O3 pollution may be reasonable. Therefore, the manuscript needs to make major revisions before their paper is considered acceptable. Please see the following comments.

**Response:** We thank the referee #2 for the constructive comments and suggestions, which are very helpful for improving the clarity and reliability of the manuscript. We have revised some sentences for better readability and enhanced the clarity of some figures. Please see our point-by-point responses to your comments below.

**Main comments:**

1. As show in manuscript, the regional O3 pollution usually occurred before and after tropical cyclones in 2018-2022 as shown in their analysis. They use the data reconstruction about the TC weather patterns to evaluate the annual change of O3 concentrations in the future, instead of extreme O3 pollution? I believe that the evaluation of extreme O3 pollution may be reasonable. Please discuss this point.

**Response:** Thank you for your careful review of our manuscript and your scientific suggestions. Based on your suggestion, we have adjusted the study period. Our previous analysis shows that O3 pollution in the Yangtze River Delta YRD typically occurs during **June–September**. This period not only coincides with the peak tropical cyclone (TC) activity but also corresponds to the highest O3 pollution occurrence, making it the

most relevant timeframe for our study. To better investigate the impact of TCs on O3 pollution, we no longer perform weather pattern classification for the entire year, but focus on June–September each year. After classifying the weather patterns, we found that the frequency and intensity of these patterns are strongly correlated with O3 concentration variations. Following previous studies, we reconstructed the annual warm season O3 variation sequence based on the frequency and intensity of the weather patterns. In this study, the reconstructed sequence also represents O3 variations during June–September each year when extreme O3 episodes occur.

We note that tropical cyclone events can lead to extreme O3 episodes. However, as our reconstruction method is based on weather pattern frequency and intensity, it does not allow for analysis of individual extreme events, although it effectively captures the main interannual variability of O3. In this study, we use this method to reconstruct future annual O3 variation sequences in warm season when extreme O3 episodes usually occur, and to explore the changes in the occurrence days of TC-related weather patterns and their contributions to the reconstructed sequences.

In our future work, we will combine numerical simulations with observational data to further investigate the relationship between tropical cyclones and extreme O3 pollution, providing a more comprehensive understanding of the impacts of TCs on O3. Now, the study is going on, and some of the reviewers' suggestions are also adopted.

Changes in the revised manuscript (lines 253–256 and 390–399) are shown as follows:

**3.2 Main synoptic weather patterns in the YRD**

"Based on the analysis of O3 variation characteristics in the YRD, O3 concentrations were generally high from **June to September**, coinciding with the peak season of TC activity. We selected June to September during 2018–2022 as the research period and used the PTT weather classification method to classify the weather situation..."

3.4.1 The role of changes in the intensity and frequency of SWPs in the reconstruction of the annual variation series of O3

"Different dominant SWPs produced varying near-surface meteorological conditions, which in turn affected atmospheric processes such as O3 photochemical

production, transport, diffusion, and wet and dry deposition. Changes in the frequency and intensity of SWPs were two key factors influencing O3 concentration variations. We reconstructed the annual time series of O3 concentrations from **June to September** between 2018 and 2022 by considering only changes in SWP frequency (fre) and changes in both SWP intensity and frequency (fre+int). In our study, we first removed the influence of emission sources on O3 concentrations based on the results of Yan et al. (2024). Subsequent analyses were conducted using O3 concentration series that were free from emission source influences. Using the O3 trend reconstruction method, we quantified the contribution of SWPs to annual variations in O3 concentrations from June to September."

2. The authors utilize the data reconstruction about the TC weather patterns to evaluate the annual change of O3 concentrations in the future. This data reconstruction of O3 concentrations only considers the effects of weather patterns, I want to know the effect from the perturbations in precursor emissions. Maybe the perturbations in precursor emissions can dominate the future change of O3 concentrations. Therefore, please check the relative contributions of climate change (or weather patterns) and perturbations in precursor emissions to future O3 concentrations.

**Response:** Thank you for your helpful suggestion. On the early stage when establishing the reconstruction relationship for weather patterns, we did not remove the influence of emissions. Considering the reviewer's comment, in the revised manuscript, we first applied detrending and **removed the effect of emission sources** from the O3 data before establishing the reconstruction relationships. This allows us to analyze the future impact of weather patterns on O3 more accurately.

We acknowledge that perturbations in precursor emissions may dominate future O3 changes, but evaluating their relative contribution is beyond the scope of the current study. The primary focus of this work is to investigate the role of tropical cyclone-related weather patterns in driving interannual O3 variability. Numerous studies have already analyzed the contribution of emissions to O3, while our study emphasizes the influence of weather patterns on future O3 trends. The reconstruction method used in

this study is also based solely on the frequency and intensity variations of weather patterns and does not include any calculation related to precursor emissions.

In future work, we will combine numerical simulations with emission scenario analyses to further quantify the relative contributions of climate/weather pattern changes and perturbations in precursor emissions to future O3 variations.

Changes in the revised manuscript (lines 390–399) are shown as follows:

3.4.1 The role of changes in the intensity and frequency of SWPs in the reconstruction of the annual variation series of O3

"Different dominant SWPs produced varying near-surface meteorological conditions, which in turn affected atmospheric processes such as O3 photochemical production, transport, diffusion, and wet and dry deposition. Changes in the frequency and intensity of SWPs were two key factors influencing O3 concentration variations. We reconstructed the annual time series of O3 concentrations from June to September between 2018 and 2022 by considering only changes in SWP frequency (fre) and changes in both SWP intensity and frequency (fre+int). In our study, we first removed the influence of emission sources on O3 concentrations based on the results of Yan et al. (2024). Subsequent analyses were conducted using O3 concentration series that were free from emission source influences. Using the O3 trend reconstruction method, we quantified the contribution of SWPs to annual variations in O3 concentrations from June to September."

3. In the global climate models, they can directly simulate the surface O3 concentrations in historical and future scenarios (e.g., Turnock et al., 2020). I suggest that the authors added the results from the Coupled Model Intercomparison Project Phase 6 (CMIP6) to further examine the Figure 9.

Turnock, S. T., Allen, R. J., Andrews, M., Bauer, S. E., Deushi, M., Emmons, L., Good, P., Horowitz, L., John, J. G., Michou, M., Nabat, P., Naik, V., Neubauer, D., O'Connor, F. M., Olivié, D., Oshima, N., Schulz, M., Sellar, A., Shim, S., Takemura, T., Tilmes, S., Tsigaridis, K., Wu, T., and Zhang, J.: Historical and future changes in air pollutants from CMIP6 models, Atmos. Chem. Phys., 20, 14547–14579,

https://doi.org/10.5194/acp-20-14547-2020, 2020.

Response: Thank you for your valuable comment. Our research reconstructs the annual ozone variation series during the warm season (June-September) based on weather patterns. Figure 9 shows the reconstruction results, which are primarily based on the frequency and intensity of weather patterns. Based on your suggestion, we conducted a comparative verification of our results. We find that the overall trends of the reconstructed O3 sequences under future scenarios are similar to those reported in previous studies (e.g., Turnock et al., 2020, Li et al., 2023), indicating that our reconstruction reasonably reflects the future trends of O3. Based on this, we revised the introduction and future reconstruction sections.

Changes in the revised manuscript (lines 95–104 and 494–508) are shown as follows:

"Affected by the surrounding atmospheric circulation, the slower the TCs move, the longer their impact duration and the greater their effects, which influence O3 transport, extend the duration of O3 pollution, exacerbate O3 concentration, and expand the spatial extent of pollution. Global climate models (GCMs) are able to directly simulate surface O3 concentrations under both historical and future scenarios (Turnock et al., 2020). These simulations provide an important reference for understanding the long-term evolution of surface O3 driven by changes in emissions and climate. In the context of global warming in the future, the increase in unfavorable meteorological conditions will make the O3 pollution problem more serious (Fu and Tai, 2015; Keeble et al., 2017; Arnold et al., 2018; Akritidis et al., 2019; Saunier et al., 2020). Therefore, studying the trend of O3 changes under future climate change scenarios is particularly important for formulating countermeasures against O3 pollution.

The average O3 concentration in the YRD from June to September during 2018 to 2022 was 123.89 μg/m³. Under the SSP2-4.5 scenario, O3 concentrations in the YRD are projected to increase relative to the historical period (Fig. 9a), with an average increase of approximately 1.88 μg/m³. Based on the reconstructed contribution of SWP changes, future O3 concentrations in the YRD are estimated to be 2.70 μg/m³ higher than in the historical period. Under the SSP5-8.5 scenario, O3 concentrations are projected to increase relative to the historical period (Fig. 9b), reaching 133.80 μg/m³

in 2100, an increase of approximately 6.86 μg/m³. Based on the reconstructed contribution of SWP changes, future O3 concentrations in the YRD are estimated to be 9.85 μg/m³ higher than in the historical period. In summary, under both the SSP2-4.5 and SSP5-8.5 future climate scenarios, O3 concentrations in the YRD are projected to increase from June to September, with more severe O3 pollution under the SSP5-8.5 scenario. Previous studies based on CMIP6 multi-model simulations have shown that surface O3 concentrations are projected to decrease in response to reductions in anthropogenic emissions, although the magnitude and spatial distribution of changes vary among scenarios (Turnock et al., 2020; Li et al., 2023). The trends revealed in this study are generally consistent with those of previous studies, lending confidence to the robustness of our findings."

**References**

Li, H., Yang, Y., Jin, J., Wang, H., Li, K., Wang, P., and Liao, H.: Climate-driven deterioration of future ozone pollution in Asia predicted by machine learning with multi-source data, Atmos. Chem. Phys., 23, 1131–1145, https://doi.org/10.5194/acp-23-1131-2023, 2023.

Turnock, S. T., Allen, R. J., Andrews, M., Bauer, S. E., Deushi, M., Emmons, L., Good,
P., Horowitz, L., John, J. G., Michou, M., Nabat, P., Naik, V., Neubauer, D.,
O'Connor, F. M., Olivié, D., Oshima, N., Schulz, M., Sellar, A., Shim, S.,
Takemura, T., Tilmes, S., Tsigaridis, K., Wu, T., and Zhang, J.: Historical and future changes in air pollutants from CMIP6 models, Atmos. Chem. Phys., 20, 14547-14579, http://doi.org/10.5194/acp-20-14547-2020, 2020.

Finally, we would like to thank the reviewers for their comments and suggestions, which improved the rigor of our work and increased the article's novelty. Once again, thank you for your time. We hope that these revisions, made based on your comments, effectively address all of your concerns regarding our manuscript.